



# Optimization of over-summer snow storage at midlatitudes and low elevation

**Hannah S. Weiss[1], Paul R. Bierman[1,2], Yves Dubief[3], and Scott D. Hamshaw[4]**

[1]Rubenstein School of Environment and Natural Resources, University of Vermont, Burlington, VT 05405, USA
[2]Geology Department, University of Vermont, Burlington, VT 05405, USA
[3]Department of Mechanical Engineering, University of Vermont, Burlington, VT 05405, USA
[4]Department of Civil & Environmental Engineering, University of Vermont, Burlington, VT 05405, USA

**Correspondence:** Hannah S. Weiss (hsweiss@uvm.edu)

**Abstract.** Climate change, including warmer winter temperatures, a shortened snowfall season, and more rain-on-snow events, threatens nordic skiing as a sport. In response, over-summer snow storage, attempted primarily using woodchips as a cover material, has been successfully employed as a climate change adaptation strategy by high-elevation and/or high-latitude ski centers in Europe and Canada. Such storage has never been attempted at a site that is both low elevation and midlatitude, and few studies have quantified storage losses repeatedly through the summer. Such data, along with tests of different cover strategies, are prerequisites to optimizing snow storage strategies. Here, we assess the rate at which the volume of two woodchip-covered snow piles (each $\sim 200\,\mathrm{m}^3$), emplaced during spring 2018 in Craftsbury, Vermont (45° N and 360 m a.s.l.), changed. We used these data to develop an optimized snow storage strategy. In 2019, we tested that strategy on a much larger, $9300\,\mathrm{m}^3$ pile. In 2018, we continually logged air-to-snow temperature gradients under different cover layers including rigid foam, open-cell foam, and woodchips both with and without an underlying insulating blanket and an overlying reflective cover. We also measured ground temperatures to a meter depth adjacent to the snow piles and used a snow tube to measure snow density. During both years, we monitored volume change over the melt season using terrestrial laser scanning every 10–14 d from spring to fall. In 2018, snow volume loss ranged from 0.29 to 2.81 $\mathrm{m}^3\,\mathrm{d}^{-1}$, with the highest rates in midsummer and lowest rates in the fall; mean rates of volumetric change were 1.24 and 1.50 $\mathrm{m}^3\,\mathrm{d}^{-1}$, 0.55 % to 0.72 % of initial pile volume per day. Snow density did increase over time, but most volume loss was the result of melting. Wet woodchips underlain by an insulating blanket and covered with a reflective sheet were the most effective cover combination for minimizing melt, likely because the aluminized surface reflected incoming short-wave radiation while the wet woodchips provided significant thermal mass, allowing much of the energy absorbed during the day to be lost by long-wave emission at night. The importance of the pile surface-area-to-volume ratio is demonstrated by 4-fold lower rates of volumetric change for the $9300\,\mathrm{m}^3$ pile emplaced in 2019; it lost < 0.16 % of its initial volume per day between April and October, retaining 60 % CE1 of the initial snow volume over summer. Together, these data demonstrate the feasibility of over-summer snow storage at midlatitudes and low elevations and suggest efficient cover strategies.

## 1 Introduction

Earth's climate is warming (Steffen et al., 2018). This warming is expressed not only in warmer nights and days but also in the number of winter rain and thaw events that degrade snowpacks (Climate Central, 2016 TS1). The duration, extent, and thickness of both lake ice and snow have decreased over the past several decades in response to increasing temperatures, especially at high latitudes (Hewitt et al., 2018; Sanders-DeMott et al., 2018). Winter recreation is particularly vulnerable to such warming. The ski industry has responded by increasing snowmaking as well as attempting to reduce melt by covering snow using various materials (Scott

and McBoyle, 2007; Pickering and Buckley, 2010; Steiger et al., 2017). Over the past several decades, ski centers have improved snowmaking strategies and facility operations both to maintain financial stability and to decrease their output of greenhouse gases (Koenig and Abegg, 1997; Moen and Fredman, 2007; Tervo, 2008; Kaján and Saarinen, 2013). Recent research focuses on analyzing and optimizing stages in the snow production cycle to assist industry efforts (Hanzer et al., 2014; Spandre et al., 2016; Grünewald and Wolfsperger, 2019).

Many sites organizing major winter sports events, such as cross-country or alpine world cup races, have adopted over-summer snow storage in response to the unpredictability of snowmaking weather conditions. In areas of high humidity and warm average fall temperatures, summer snow storage is more reliable than expecting weather conditions to be sufficiently cold and dry for making snow at the start of the winter ski season. For example, the 2014 Olympic Games at Sochi relied on $750\,000\,\mathrm{m}^3$ of stored snow (Pestereva, 2014).

Over-summer storage of snow and ice is not a new idea; for example, ice houses stored large blocks of lake ice beneath sawdust over the summer (Nagnengast, 1999; Rees, 2013). Today, the ski industry uses stored snow to support the early winter ski season. Modern over-summer snow storage (sometimes referred to as "snow farming") begins with the creation of snow piles during winter months. Piles are covered (often with sawdust or woodchips and sometimes geotextiles) before the snow is stored over the summer (Skogsberg and Lundberg, 2005). In the fall, the pile is uncovered and snow spread onto trails. Nordic ski centers require less snow-covered area to open than downhill ski centers, and so snow storage on the scale of thousands of cubic meters is practical and cost-effective, allowing the center to open on time instead of losing business, which occurs if centers are unable to make snow and thus must open later. Snow storage has been employed predominantly at high-elevation and/or high-latitude ski centers (Fig. 1), many of which benefit from cool, dry summers that minimize energy transfer to the snow, increase evaporative cooling, and thus slow snowmelt.

Here, we examine the feasibility of snow storage in the northern United States at a midlatitude, low-elevation ($45°$ N and $360\,\mathrm{m}$ a.s.l.) site with a humid, temperate climate, including warm summer temperatures and high relative humidity which limits evaporative cooling (Fig. 1). Out of the 28 known snow storage locations, our study location has the highest average June–July–August temperature ($24\,°\mathrm{C}$) and highest solar-radiation levels (Worldclim – Global Climate Data, http://worldclim.org/version2, last access: 14 September 2019)TS2 TS3. In this paper, we report data on the rate of volumetric change of snow stored over the summer and consider those data in the context of both ground temperature and meteorological data that together help define the energy flux, which is responsible for melt into and out of the snow piles. The goals of this research are to (1) determine the rate of volumetric change of small experimental snow piles, (2) suggest an optimized snow storage strategy based on those data, and (3) test the optimized strategy on a larger snow pile sufficient for ski area opening. Our data fill a research gap in measurements of volumetric change during snow storage and provide a novel case study for snow storage at low-elevation and midlatitude sites.

## 2   Background

Although the physics of snowmelt has been considered extensively (Dunne and Leopold, 1978; Horne and Kavaas, 1997; Jin et al., 1999), there has been limited application of physical and energy transfer knowledge to the problem of over-summer snow storage (Grünewald et al., 2018). Snowmelt occurs when the snowpack absorbs enough energy to raise snow temperature to the melting point ($0\,°\mathrm{C}$) and then absorbs additional energy to enable the phase change from solid to liquid water ($0.334\,\mathrm{MJ\,kg}^{-1}$). The snowpack gains energy from incoming short- and long-wave radiation, sensible and latent heat transfer from condensation of atmospheric water vapor and cooling and refreezing of rainwater, conduction from the underlying ground, and advective heat transfer from wind (Dunne and Leopold, 1978). Loss of energy from the snowpack occurs through convective and conductive heat transfer to the air, evaporative cooling, and long-wave emission to the atmosphere.

Both regional and local climatic factors influence the energy balance of snow. Short-wave radiational gain is related to latitude (highest near the Equator and least near the poles), elevation, time of year (greatest in summer and least in winter), snow pile surface albedo, slope and aspect, and cloud and tree canopy cover. Long-wave radiation balance depends on atmospheric emissivity, cloudiness, vegetation cover, and temperature of the snow pile surface. Rain falling on the snowpack transfers heat. Conductive heat transfer from the ground depends on soil thermal conductivity and temperature (Kane et al., 2000; Abu-Hamdeh, 2003). Snowmelt typically varies on a diurnal cycle, with melt increasing after sunrise, peaking in the afternoon, and decreasing after sunset (Granger and Male, 1978). Once surface melt occurs, water either refreezes if it percolates into a sub-freezing snowpack, flows through an isothermal ($0\,°\mathrm{C}$) snowpack and then infiltrates into the ground below, or flows along the ground surface below the pile, depending on the soil infiltration rate (Schneebeli, 1995; Ashcraft and Long, 2005).

Recent research at nordic ski centers in Davos, Switzerland, and Martell, Italy (Grünewald et al., 2018), has applied snowmelt physics to optimize over-summer snow storage at high-elevation ($\sim 1600\,\mathrm{m}$) and midlatitude ($\sim 46°$ N) sites. The Davos location has an average summer relative humidity of $79\,\%$. Each nordic center built piles of machine-made snow and covered them with $40\,\mathrm{cm}$ of wet sawdust and woodchips; researchers then used utilized terrestrial laser scanning to measure the initial (spring) and final (fall) vol-

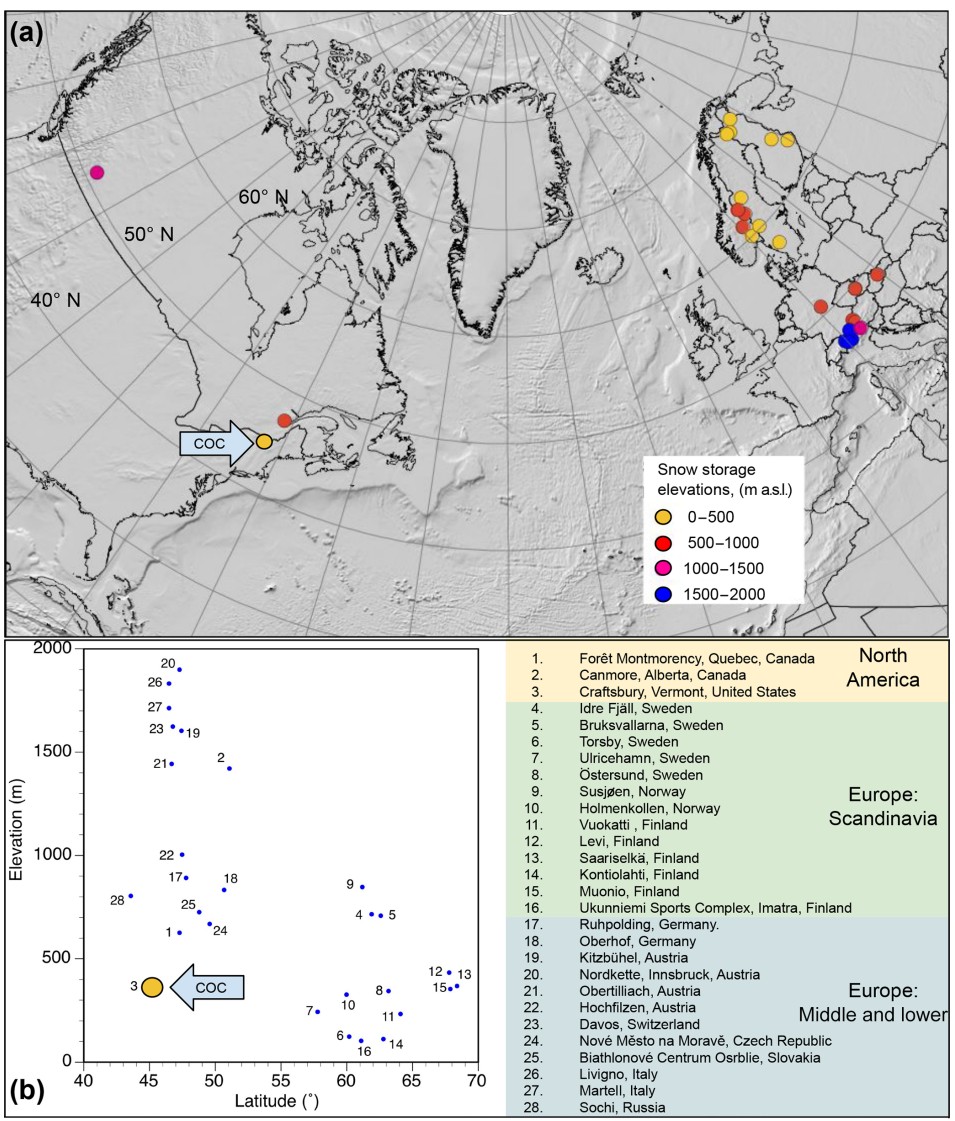

**Figure 1.** Locations of known over-summer snow storage sites (both currently active and inactive). **(a)** Conical projection shows known locations of over-summer snow storage at nordic ski centers. The Craftsbury Outdoor Center is highlighted with a blue arrow, which is labeled COC. The relative elevations of ski centers are displayed as a color gradient, marked in the legend. **(b)** Scatterplot of same locations as shown in **(a)**. The Craftsbury Outdoor Center (no. 3) is large yellow dot (COC). It has the lowest combination of elevation and latitude of any snow storage yet attempted.

umes of the two piles. These snow piles retained 74 % and 63 % of their volume over the summer. Using a physically based model, Grünewald et al. (2018) suggested that the most effective cover, in relation to work and cost, was a 40 cm thick layer of mixed wet sawdust and woodchips, which reduced energy input into the pile by a factor of 12 (1504 MJ m$^{-2}$ without woodchips as opposed to 128 MJ m$^{-2}$ with woodchips). Deeper cover layers can save more snow, but costs are higher. During the day, solar radiation caused evaporation from surface woodchips while capillary flow continually supplied moisture from the melting snow to the surface. The wet woodchips and sawdust also provided ther-

mal mass, slowing the transfer of energy from the surface to the snow beneath.

Lintzén and Knutsson (2018) reviewed current knowledge of snow storage and experience from areas in Scandinavia and reported new results from an experiment in northern Sweden, analyzing melt loss of stored snow. They report that the most common snow storage method employs a breathable surface layer over an insulating material. From field observations at multiple nordic ski centers, they have found that the choice and age of covering affects the melt rate; older woodchips were less effective at reducing melt than fresh chips. Lintzén and Knutsson also determined that woodchips

were a more effective cover than bark. They measured snow volumes three times over the summer and found that higher relative humidity increased the melt rate. They also investigated the geometry of snow piles and determined that shaping piles, in a way that maximized the ratio of volume to surface area, minimized melt loss; however, steeper snow pile sides caused sliding and failure of cover materials (Lintzén and Knutsson, 2018).

Data related to snow storage for the purpose of summer cooling to improve energy efficiency and comfort supplements those gathered from ski centers. In central Sweden, the Sundsvall Hospital conserves snow over the summer for air conditioning with a $140\,\mathrm{m} \times 60\,\mathrm{m}$ storage area (holding $60\,000\,\mathrm{m}^3$ snow) underlain by watertight asphalt (Nordell and Skogsberg, 2000). After covering with $20\,\mathrm{cm}$ of woodchips, the majority of natural snowmelt resulted from heat transfer from air (83 %), while heat transfer from groundwater drove 13 % of melt and heat from rain accounted for 4 % of melt. Similar work was done by Kumar et al. (2016) and Morofsky (1982) in Canada and by Hamada et al. (2010) in Japan.

## 3    Methods and setting

### 3.1    Study location

We conducted our experiment at the Craftsbury Outdoor Center (COC), a sustainability-focused, full-year recreation venue located in northeastern Vermont at $360\,\mathrm{m}$ a.s.l. (Fig. 1), an area with warm, humid summers and cold, dry winters. The COC maintains $105\,\mathrm{km}$ of groomed nordic ski trails and hosts national and international races several times each winter. Average maximum monthly air temperature at St. Johnsbury, Vermont (closest National Oceanic and Atmospheric Administration – NOAA – station to the COC about $30\,\mathrm{km}$ southeast; at $215\,\mathrm{m}$ a.s.l.), between 1895 and 2018 ranges between $3.6\,°\mathrm{C}$ (January) and $29\,°\mathrm{C}$ (July), mean temperature ranges from $-8.3\,°\mathrm{C}$ (January) to $20.7\,°\mathrm{C}$ (July), and minimum air temperature ranges between $-34\,°\mathrm{C}$ (December) and $15\,°\mathrm{C}$ (July, Climate Summary for Saint Johnsbury, VT, https://www.fairbanksmuseum.org/eye-on-the-sky/summaries-for-st-js-climate/normals-and-extremes, last access: 6 February 2019) TS4 . Soils in the area are very rocky, silty loam, sandy loam, and loam developed on glacial till (Web Soil Survey, https://websoilsurvey.nrcs.usda.gov/app/WebSoilSurvey.aspx, last access: 20 October 2018) TS5 . Average summer precipitation is $\sim 300\,\mathrm{mm}$ (NOAA, 2019). The most common land-cover types are forest and woodlands (USGS, https://mrdata.usgs.gov/geology/state/fips-unit.php?code=f50019, last access: 15 October 2018) TS6 .

### 3.2    Initial snow pile experiments

On 30 March 2018, two snow piles were emplaced at the COC using PistenBully snow groomers at two separate sites (Fig. 2). Site 1 is adjacent to the COC's main campus buildings in direct sunlight, with minimal wind protection. Site 2 is $1\,\mathrm{km}$ north of Site 1, within a cleared depression in the forest which also in direct sunlight but more protected from wind than Site 1. At the time of emplacement, the snow was transformed and had a density of $> 500\,\mathrm{kg}\,\mathrm{m}^{-3}$ (see Sect. 3.5 for snow density measurement methods). At Site 1, $225\,\mathrm{m}^3$ of machine-made snow was banked against a north-facing slope. At Site 2, $210\,\mathrm{m}^3$ of natural snow was shaped into a symmetrical, rounded pile. The two piles were draped with thin sheets of clear plastic. The plastic sheets, about $0.15\,\mathrm{mm}$ thick, were impermeable and emplaced to prevent woodchips from mixing with the snow. The piles were then covered with an irregular layer of woodchips averaging $20 \pm 10\,\mathrm{cm}$ (1 SD) on 21 April 2018; chip thickness ranged from a minimum of $6\,\mathrm{cm}$ to a maximum of $40\,\mathrm{cm}$ (Fig. 3). In early July, about $50\,\mathrm{m}^3$ of snow were removed from the pile at Site 1 by COC personnel, the plastic was removed, and the remaining snow was covered again with woodchips and left for continued monitoring.

### 3.3    Weather stations

Weather stations adjacent to each pile and $3–4\,\mathrm{m}$ above the ground surface (Davis Vantage Pro2) collected air temperature, humidity, precipitation, solar-radiation, wind speed and direction, and barometric-pressure data. The weather stations record data at $15\,\mathrm{min}$ intervals and transfer them to the Web, where they are publicly accessible (https://wunderground.com/personal-weather-station/dashboard?ID=KVTCRAFT2#history, last access: 23 October 2019). Local soil temperature was measured with temperature sensors installed at four depths within the soil (5, 20, 50, and 100 or $105\,\mathrm{cm}$ below the surface) adjacent to each snow pile. Two HOBO Onset data loggers recorded temperatures at four depths at $20\,\mathrm{min}$ intervals between June 2017 and October 2018.

### 3.4    Terrestrial-laser-scanning field methods and processing

CE2 During spring and summer, the shape and volume of the piles were measured every $10–14\,\mathrm{d}$ using a terrestrial laser scanner (RIEGL VZ-1000). Terrestrial laser scanning (TLS) is an accurate method for obtaining digital surface models (DSMs) of various terrain types, including snow surfaces (Prokop et al., 2008; Molina et al., 2014). Six to ten permanent tie points around each pile were established during the initial survey by fastening reflective $5\,\mathrm{cm}$ disks to stable surfaces such as large trees and buildings. The first survey was done prior to snow pile placement in order to establish

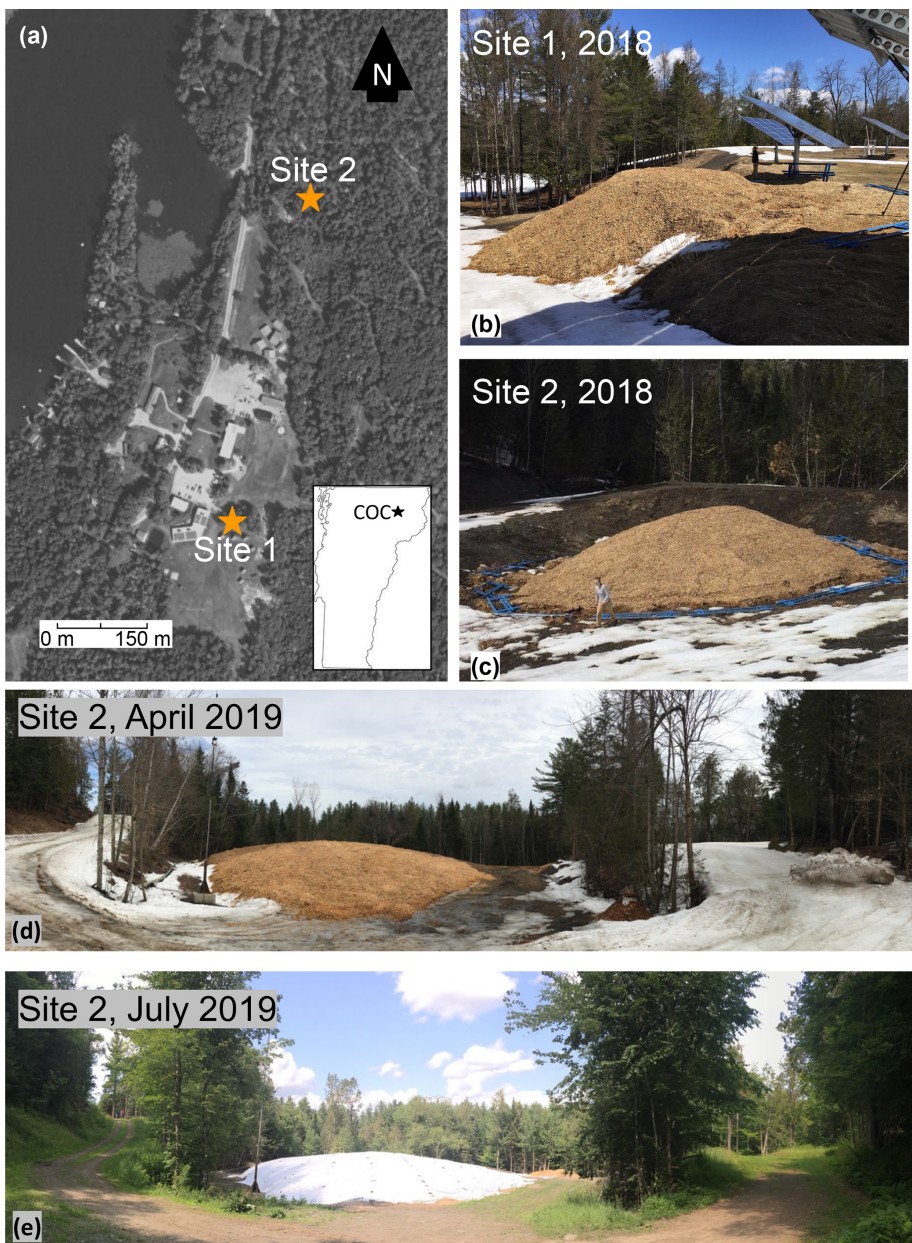

**Figure 2.** Snow storage at Craftsbury Outdoor Center. **(a)** Aerial view of the Craftsbury Outdoor Center (COC) in Vermont, from http://maps.vcgi.vermont.gov (8 February 2019). Both study site locations shown by number. **(b)** Site 1 (225 m$^3$), covered in woodchips on 21 April 2018, with trees and solar panels for scale. **(c)** Site 2 (209 m$^3$) when installed. Site 1 received 24 m$^3$ of woodchips, and Site 2 received 42 m$^3$ of woodchips. Person for scale. **(d)** Site 2 in April 2019; 9300 m$^3$ of snow, eventually covered with 650 m$^3$ of woodchips. **(e)** Site 2 in July 2019, the snow pile overlain by a reflective geofabric. Trees for scale.

ground surface topography. Tie-point locations were determined and fixed relative to the scanner GPS position during the initial scan. Each survey consisted of three or four scans per site (depending on available vantage points), which were combined in the RiSCAN Pro software version 2.6.2 (RIEGL Laser Measurement Systems GmbH: RiScan Pro, 2011). Scan registration was done in RiSCAN using a combination of tie-point registration (finding corresponding points) and the multi-station adjustment routine using plane patches and tie objects. Similar studies of monitoring bare and covered snow surfaces with TLS have applied this technique (Prokop et al., 2008; Grünewald et al., 2018; Grünewald and Wolfsperger, 2019). Scans were collected at a horizontal and vertical angular resolution of 0.08°. Scans were collected from distances less than 100 m, resulting in average point spacing over the pile < 1 cm.

To calculate snow pile volumes and volumetric change over time (between scans), point clouds of each pile were processed into DSMs. Processing the workflow involved cropping the point cloud to the area of interest in RiSCAN Pro and exporting cropped point clouds into LAS format, projected into Vermont State Plane NAD83 coordinates. Point clouds were converted to a 10 cm resolution DSM using the min-Z filter and QT Modeler software (version 8.0.7.2) and adaptive triangulation to fill in small data gaps. Volume calculations and differences in volume between sequential surveys were calculated in QT Modeler using these DSMs.

### 3.5 Density

Snow density was measured using a Rickly Federal Snow Sampling Tube. The snow tube was weighed, pushed into the snow, removed, and weighed again. The weight of the tube was subtracted from the combined weight of the snow and tube, and density was calculated by dividing the mass of snow by its volume (length of snow within the tube multiplied by the area of the opening; $\sim 13\,\mathrm{cm}^2$). Density was collected three times (in March, May, and July) at the top surface of pile 1 during 2018. In 2019, density was collected once at the top of the pile in February.

### 3.6 Cover experiments

Cover experiments were performed at both sites in June and July 2018. At Site 1, two 5 cm thick, impermeable, rigid foam boards ($R = 3.9$ per 2.5 cm; value expressing resistance to conductive heat flow) were stacked and compared to a 20 cm, uniform, porous layer of woodchips both with and without a reflective cover (aluminized space blanket). At Site 2, we covered snow with a double-layered, 2.5 cm thick insulating concrete curing blanket ($R = 3.3$ per 2.5 cm) and overlaid the blanket with either open-cell, permeable foam ($R = 3.5$ per 2.5 cm) or a uniform, porous layer of woodchips (20 cm thickness), both with a reflective cover. For both foam experiments, woodchips and plastic sheeting were removed from the test area. For woodchip experiments, plastic sheeting was removed from the test area. Individual cover experiments were conducted in areas of 1 m² each, with thermosensors placed in the center of each quadrat at varying depths between layers (Table 2; Fig. 4).

### 3.7 Power spectral density function

We computed the power spectral density (PSD) function to determine relative effectiveness of the different covers. The temperature signal is first decomposed in a series of waves of well-defined frequencies:

$$T(t) = \frac{1}{N} \sum\nolimits_{k=0}^{N-1} \hat{T}_k \exp\left(i 2\pi f_k t\right), \tag{1}$$

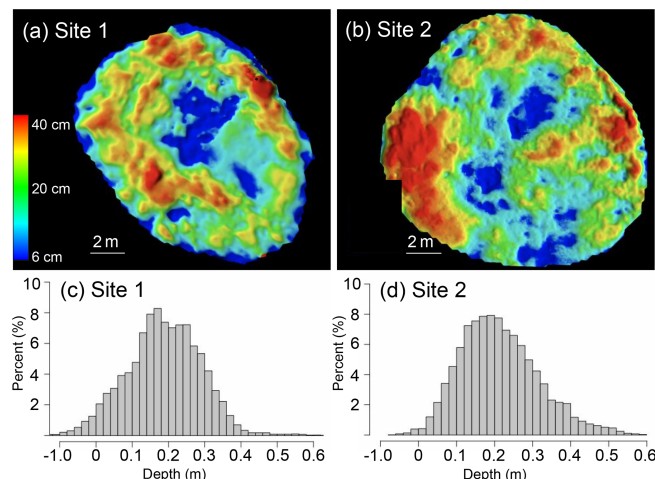

**Figure 3.** Woodchip thickness distribution maps of pile 1 **(a)** and pile 2 **(b)**, with red indicating areas of high thickness and blue indicating areas of low thickness. Panel **(c)** represents the chip thickness histogram for pile 1, and **(d)** is chip thickness histogram for pile 2. Negative thickness values likely represent snow settling between bare-snow survey and survey after woodchip emplacement.

where $\hat{T}_k$ is the Fourier mode at frequency $f_k = k/2\Delta T$, $1/\Delta T$ is the sampling frequency of temperature acquisition, and $N$ is the number of samples in the time series. The Fourier mode contains both amplitude and phase information for each wave. The PSD is the power of the signal,

$$\mathrm{PSD}(T) = \frac{\Delta t}{N} \sum\nolimits_{k=0}^{N-1} \left| \hat{T}_k \right|^2, \tag{2}$$

which is the sum of the contributions of each wave to the power (or variance) of the signal. Typically plotted on a log–log plot, the norm of the Fourier modes as a function of frequencies is a powerful tool for detecting dominant frequencies (Welch, 1967). In the summer, the dominant oscillation in temperature is diurnal; thus, using PSD, we can judge the effectiveness of cover materials by their ability to damp the diurnal temperature signal and relevant harmonics. We computed the PSD for all temperature records in selected cover experiments (Fig. 4b, e, f).

### 3.8 Validating cover method, summer 2019

Based on data collected during summer 2018, the COC chose Site 2 (Fig. 2) as their snow storage site for 2019. Cost and ease of installation mandated a two-layer cover system – a $\sim 30$ cm thick layer of woodchips capped with a reflective, permeable covering. No plastic was placed between the woodchips and the underlying snow. The 2019 snow pile filled a drained, oblong pond basin and was gently sloped. During February, machine-made snow was blown into the pile using fanless snowmaking wands. Snow density at and just after emplacement was high, ranging between 500 and 600 kg m$^{-3}$. In March, the snow pile was shaped and fur-

**Table 1.** Weather parameters measured between June 2017 and October 2018 at the Craftsbury Outdoor Center, Craftsbury, VT.

|  | Air temperature (°C) | Relative humidity (%) | Precipitation (mm d$^{-1}$) | Solar radiation (W m$^{-2}$) |
|---|---|---|---|---|
| Minimum | −28 | 14 | 0 | 0 |
| Maximum | 33 | 93 | 22 | 1144 |
| Mean | 9 | 79 | 0.1 | 109 |
| Standard deviation | 12 | 15 | 0.4 | 205 |

**Table 2.** Properties of the 2018 covering experiments.

| Plot reference Fig. 4 | Site number* | Snow-interface layer | Middle layer | Top layer |
|---|---|---|---|---|
| (a) | 1 | None | 20 cm layer of woodchips | None |
| (b) | 1 | None | 20 cm layer of woodchips | Reflective covering |
| (c) | 1 | None | Two stacked rigid foam boards | None |
| (d) | 1 | None | Two stacked rigid foam boards | Reflective covering |
| (e) | 2 | Concrete curing blanket | 20 cm layer of woodchips | Reflective covering |
| (f) | 2 | Concrete curing blanket | 20 cm layer of open-cell foam | Reflective covering |

* The experiment at Site 1 occurred in June 2018, while the experiment at Site 2 occurred in July 2018.

ther compacted with PistenBully groomers and excavators; at that time, TLS showed that the pile had a volume of about 9300 m$^3$ without woodchips. During the next 6 weeks, the snow pile was allowed to compact and grow denser. In late April, most of the pile was covered in woodchips. By the end of May, additional woodchips were obtained and snow pile covering was completed (total woodchip volume ∼ 650 m$^3$). Using the exposed surface area of the pile without woodchips (2300 m$^2$) and the volume of woodchips, we calculate that the average woodchip thickness was 28 cm. By the end of June, the snow pile was covered in a white, 75 % reflective, breathable Beltech 2911 geofabric, secured by ropes and rocks to prevent wind disruption. Between March and October, the pile was repeatedly scanned using TLS; data were processed using methods described in Sect. 3.4.

## 4   Results

### 4.1   Meteorological data and ground temperature data

Climate at the COC is strongly seasonal – such seasonality is clear in the meteorological data collected between June 2017 and October 2018 (Fig. 5). Between June 2017 and October 2018, air temperature varied between −28.2 and 33 °C (mean annual temperature of 6 °C). Precipitation fell at a maximum rate of 22 mm d$^{-1}$ (mean of 0.06 mm d$^{-1}$), and relative humidity ranged between 14 % and 93 % (mean of 78 ± 15 %). Solar radiation had a 24 h average of 109 W m$^{-2}$ and maximum of 1144 W m$^{-2}$ (Table 1). Air temperature and solar radiation followed similar trends over the 16 months, decreasing during winter months

and increasing during summer months. Precipitation did not follow any significant pattern, and relative humidity remained high (NOAA classifies above 65 % as high, and relative humidity remained above this level for the summer), varying more during summer than winter months. Average summer temperature in 2018 (June, July, and August 2018; 22.4 °C) was ranked by NOAA as "Much above the average of 20.7 °C"; in 2019, average summer temperature ranked "above average" (21 °C). Both years had near-average precipitation (National Oceanic and Atmospheric Administration Forecast Office, Burlington VT, 2018; Craftsbury Outdoor Center KVTCRAFT2, https://www.wunderground.com/personal-weather-station/dashboard?ID=KVTCRAFT2#history, last access: 12 December 2018).

Ground temperature from all four depths at both locations followed similar trends. The shallowest sensor (5 cm below the surface) recorded the greatest variance over time (SD = 7.4 °C for Site 1). Ground temperature variations decreased in amplitude as soil depth increased; at 1 m in depth, the atmospheric temperature signal was damped (SD = 3.9 °C for Site 1). Ground temperatures for all depths showed consistent warming from installation (11 June 2017) through late August 2017 and then decreased through February 2018. The shallowest sensor revealed slight warming after February, while the deeper sensors remained stable until May 2018. During May, warming increased more noticeably for all four sensors. Ground temperature depth trends inverted during both May and November. During the winter, the coldest temperatures were at the surface; during summer, the coldest temperatures were at depth. Figure 5 displays data from sensors adjacent to pile 1 – data were collected at both sites

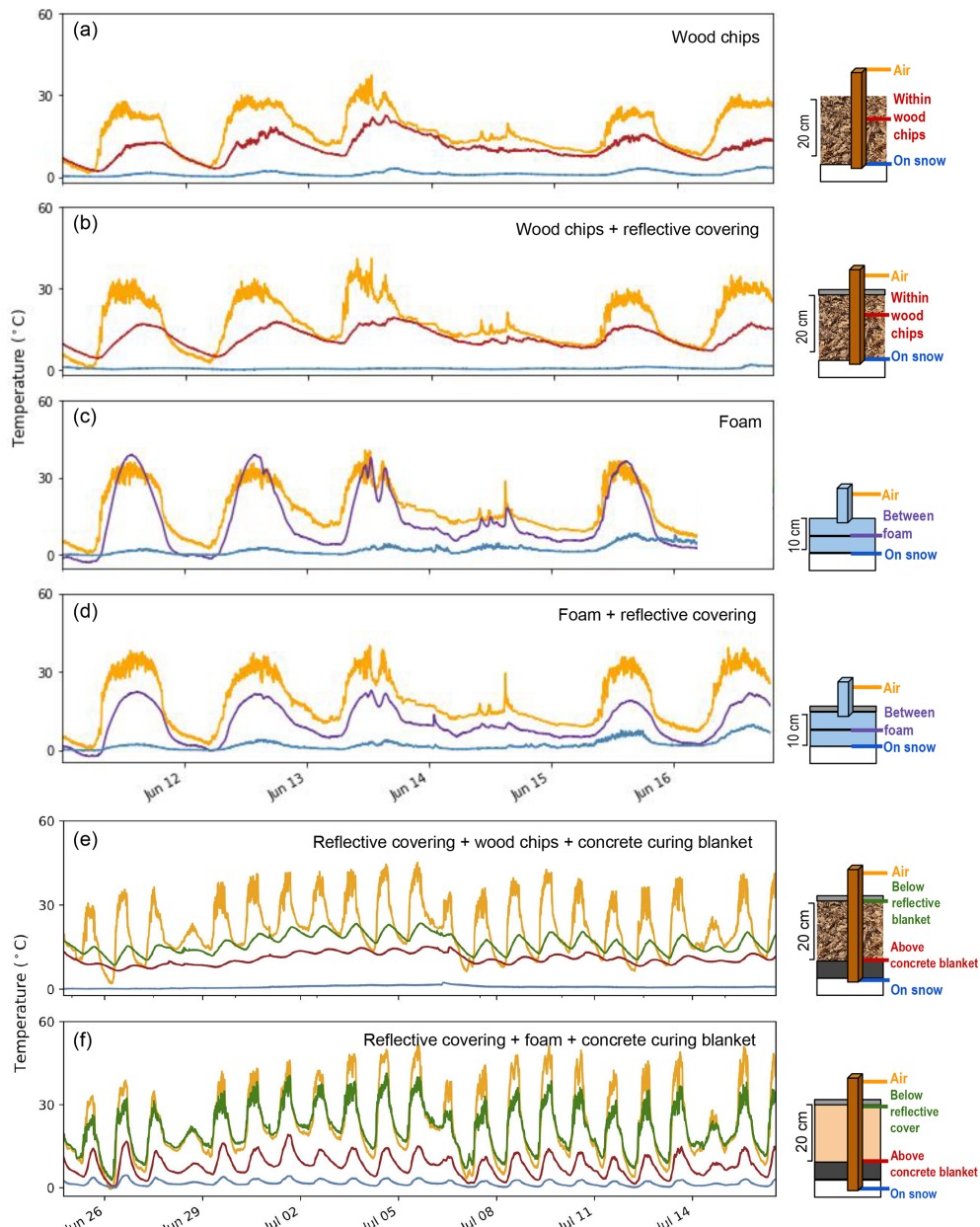

**Figure 4.** Cover experiments and resulting temperature records. **(a)** Site 1 – woodchips. **(b)** Site 1 – woodchips overlain by reflective cover. **(c)** Site 1 – foam. **(d)** Site 1 – foam overlain by reflective cover. **(e)** Site 2 – woodchips underlain by concrete curing blanket and overlain by reflective cover. **(f)** Site 2 – open-cell foam underlain by concrete curing blanket and overlain by reflective cover.

but are missing from Site 2 between 12 December 2017 and 21 April 2018.

## 4.2  Snow volume and density

Snow in both 2018 piles lasted until mid-September; however, snow volume decreased consistently throughout the summer (Figs. 6 and 7). Comparing the laser-scan survey completed just after woodchip emplacement with the initial bare snow survey showed that the layer of chips ranged in depth from 6 to 40 cm, with an average of $19 \pm 11$ cm for pile 1 and $21 \pm 11$ cm (1 SD) for pile 2 (Fig. 3). After the addition of woodchips, snow volume in both piles decreased following similar trends (Fig. 7); initial decreases in volume were partly related to compaction and increases in snow density, as snow density was $\sim 500\,\mathrm{kg\,m^{-3}}$ at emplacement, $600\,\mathrm{kg\,m^{-3}}$ in May, and $700\,\mathrm{kg\,m^{-3}}$ in July. Relative to newly fallen snow ($100-200\,\mathrm{kg\,m^{-3}}$), the snow in these piles was closer in density to ice ($900\,\mathrm{kg\,m^{-3}}$). These measurements are supported by qualitative observations of

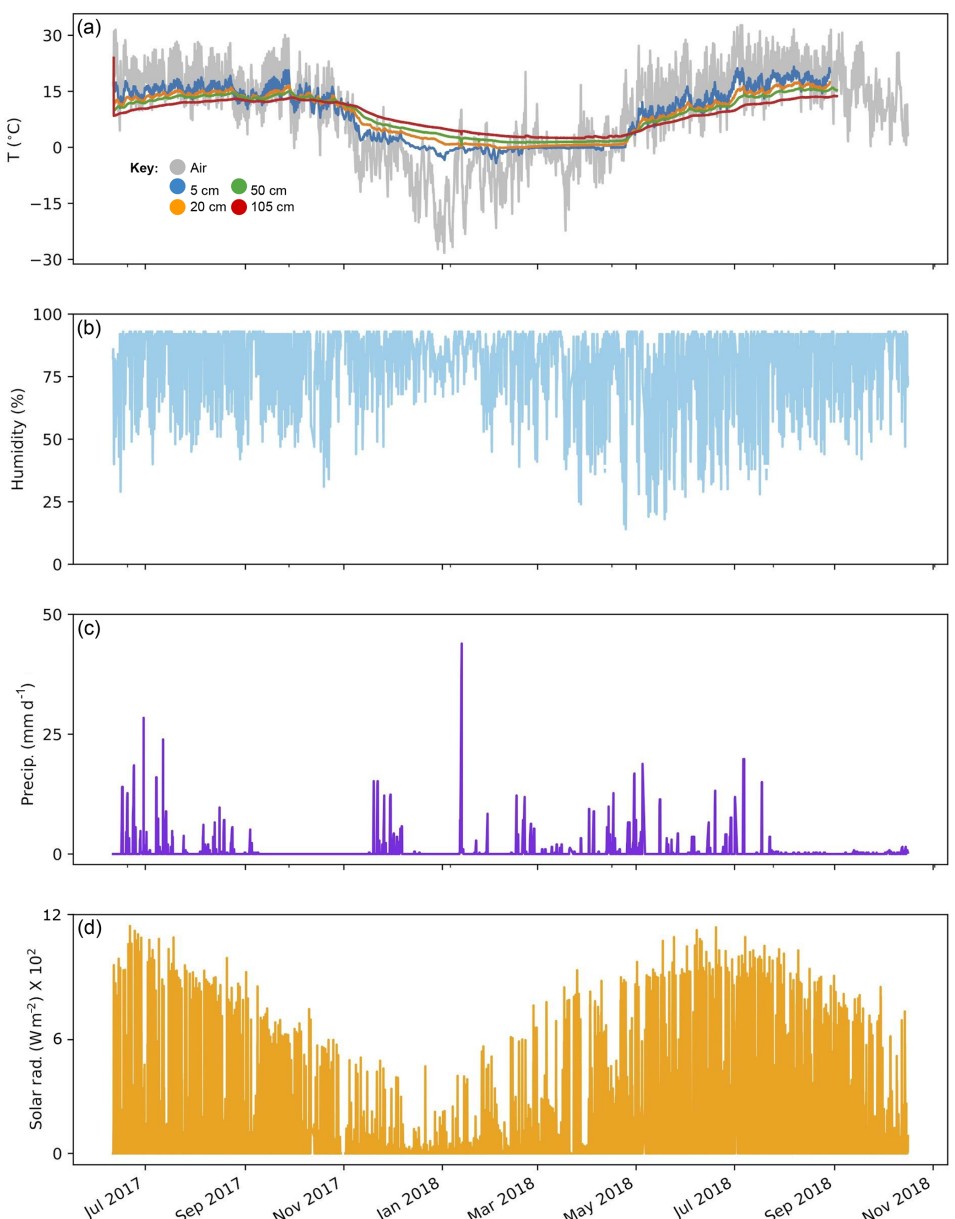

**Figure 5.** Meteorological conditions and soil temperature between 11 June 2017 and 16 October 2018. Weather conditions were collected by a Davis weather station at the Craftsbury Outdoor Center near Site 2. **(a)** Air temperature (grey), collected at 30 min intervals plotted with ground temperatures. Ground temperatures were collected at 20 min intervals adjacent to Site 1 by four HOBO Onset data loggers at depths below the ground surface of 5 cm (blue), 10 cm (orange), 50 cm (green), and 105 cm (red). Ground temperature record ends on 2 September 2018. **(b)** Relative humidity (%). **(c)** Precipitation (mm d$^{-1}$). **(d)** Solar radiation (W m$^{-2}$).

changes in snow crystal morphology over the summer (increased rounding), increasing size (up to 5 mm by July), wetness (higher liquid water content), and clarity (from white to clear by summer's end). Continued volume loss over the summer was predominately the result of melt. Average rates of volume change for both piles were relatively similar (1.24 and 1.50 m$^3$ d$^{-1}$), representing 0.55 % to 0.72 % of initial pile volume per day. Maximum loss rates, recorded in July, reached 1.98 and 2.81 m$^3$ d$^{-1}$ (Fig. 7.) As summer shifted into fall, the loss rate decreased (Fig. 7). Minimum rates of change for both piles occurred in September and were 0.29 and 0.88 m$^3$ d$^{-1}$.

As the piles decreased in volume over the summer, crevasses formed along the edge of the plastic sheeting, which exposed the snow to direct sunlight and thus increased rates of volumetric change (Fig. 6). On pile 1, a crevice formed from east to west where the pile began to slope downward (Fig. 6b). Slope failure was a potential catalyst for the

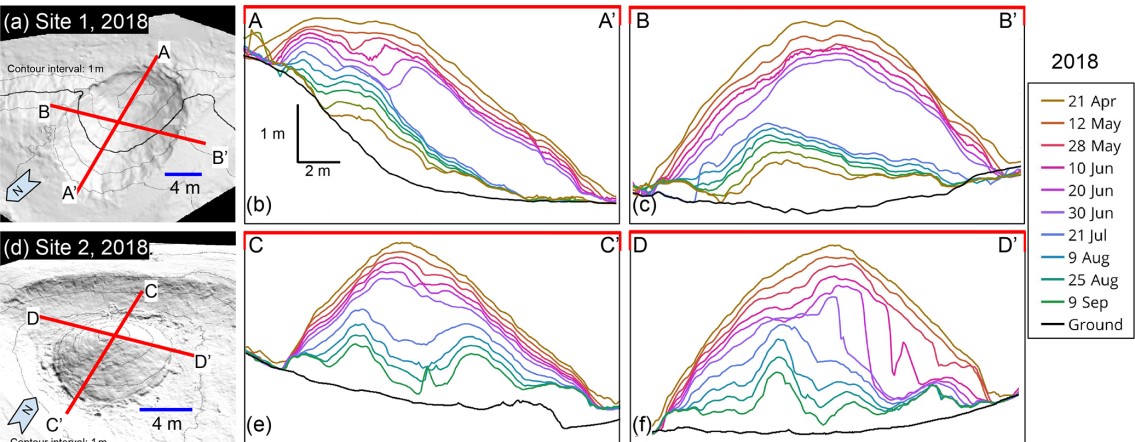

**Figure 6.** Snow pile topographic change over time in 2018. **(a)** Oblique view of digital surface model (1 m contours) of 2018 snow pile at Site 1 with cross sections $A$–$A'$ and $B$–$B'$ (21 April 2018). **(b)** Profiles for each terrestrial-laser-scan survey (21 April to 9 September 2018; $n = 13$) along section $A$–$A'$. **(c)** Profiles for each survey along section $B$–$B'$. On 3 July 2018, 30 m$^3$ of snow was removed CE5 from the pile at Site 1. **(d)** Oblique view of digital surface model (1 m contours) of 2018 snow pile at Site 2 with cross sections $C$–$C'$ and $D$–$D'$ (21 April 2018). **(e)** Profiles for each terrestrial-laser-scan survey (21 April to 9 September 2018; $n = 12$) along section $C$–$C'$. **(f)** Profiles for each survey along section $D$–$D'$. Each scan represented by a line in panels **(b)**, **(c)**, **(e)**, and **(f)** as indicated in key.

formation of crevices. We did not observe meltwater around either of the piles, suggesting that melt occurred at a rate which allowed for infiltration into the rocky sandy loam soil below. The woodchips deeper in the cover remained cold and wet throughout the summer, while the woodchips on the surface were consistently dry in the absence of rainfall.

### 4.3 Cover experiments

Thermal buffering is a function of air temperature, longwave emissions, and turbulent fluxes. We chose temperature at the snow–cover interface to indicate cover efficiency because all experiments were subjected to similar external conditions and because we have continuous data series of temperature in, above, and below the cover during each of the experiments. Two experiments preformed on 1 m$^2$ plots on each snow pile revealed that different combinations of cover materials resulted in a variety of cover efficiencies (Fig. 4). Each experiment lasted between about 1 and 3 weeks and took place in June and July, respectively. We assessed cover efficiency by determining which material combination maintained the lowest and steadiest temperature at the snow–cover interface and which most effectively damped the diurnal temperature signal (detected using PSD analysis). On the rigid foam, open-celled foam, and woodchip plots, the highest temperature was measured in the air above the surface (max of 41.2 °C; Fig. 4f). During the first experiment, air temperatures above the reflective blanket were higher than above the non-reflective surface. When all plots were covered with a reflective blanket, all air temperatures above the pile were similar; however, temperatures at lower depths, and under different cover materials (woodchips and open-cell foam), varied significantly. The lowest and most stable temperatures at the

snow–cover interface resulted when the stored snow was covered directly with an insulating concrete curing blanket, then with 20 cm of wet woodchips, and finally with a reflective sheet.

### 4.4 Power spectral density

PSD analysis provides insight into the dynamics of heat transfer in the snow piles. Figure 8 shows the log–log plot of temperature power spectral densities for three different cover experiments. It is important to realize that (i) each line represents the PSD at specific distance from the snow surface, (ii) that the integral under each line is equal to the standard deviation of the signal, or the energy of the signal fluctuations, and (iii) that the horizontal axis is frequency, thereby breaking down the total energy of the temperature signals into the individual contributions of each frequency involved in the PSD. Furthermore, the frequency is normalized by the frequency of 1 d or diurnal frequency $f_{\text{diurnal}} = 1/(24 \times 3600)$. Consequently, the horizontal coordinates 1, 2, and 4 are the diurnal (1 per 24 h), half-diurnal (1 per 12 h), and quarter-diurnal (1 per 6 h) frequencies, with 2 and 4 being harmonics of the diurnal frequency. These frequencies are highlighted by the peaks in the PSD of temperature outside of the pile (the air $T$ sensor at 46 cm). The PSD values at these frequencies are much higher than the values at surrounding frequencies, indicating that their contribution to the total energy of the signal, and therefore to the dynamics of heat transfer, is significant.

Detection of diurnal temperature swings and their harmonics in temperature records collected at different depths in the cover materials with various relative strengths is critical to understanding how cover materials minimize heat trans-

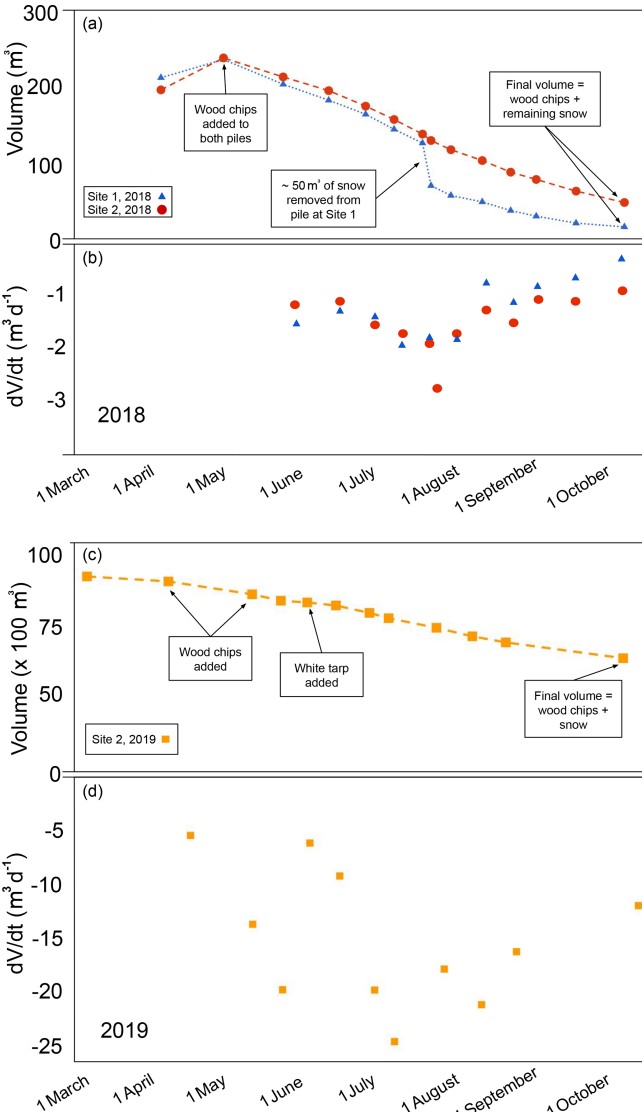

**Figure 7.** Volume change over time for snow piles at sites 1 and 2 measured by terrestrial laser scanning. **(a)** Volume of snow piles from placement in March 2018 until September 2018. Addition of woodchips in April and removal of snow in July at pile 1 shown by black arrows. Volumes are total, including woodchips. **(b)** Change in volume per unit time between surveys. The rate of volume loss increases midsummer for both piles. Site 1 received about 24 m$^3$ of woodchips, while Site 2 received about 42 m$^3$ of woodchips – this difference is due to pile geometry and the resulting difference in surface area. Site 1 snow was banked against the side of a hill, while the Site 2 pile was a hemisphere in the middle of an open depression. **(c)** Volumes of snow pile (2019) beginning in March and ending in October. Addition of woodchips throughout May and addition of white tarp are indicated by black arrows. Volumes include woodchip volume. **(d)** Change in volume per unit time between surveys.

fer. In the foam cover experiment (Fig. 8c), the diurnal frequency and its harmonics are detectable in all layers; however, the three-layer system (insulating blanket, wet woodchips, and reflective cover; Fig. 8b) fully damps all oscillations, as shown by the flatness of the PSD below the cover (0 cm; snow $T$ sensor; thick blue line). In the absence of an insulating blanket, the two-material cover system (reflective cover and woodchips) is slightly less efficient at damping the diurnal oscillation (Fig. 8a). In the case of foam, the dynamics of heat transfer at the surface, or cyclic events that drive fluctuations of temperature, are directly and efficiently transmitted to the snow surface. Such a response can be modeled as quasi-steady heat transfer conduction, which is not surprising for an inorganic dry material.

Woodchips profoundly affect the dynamics of heat transfer, and in the most dramatic case (Fig. 8b), the snow surface temperature appears to be insensitive to the diurnal and harmonic frequencies of atmospheric temperature. This indicates that the system can no longer be modeled under quasi-steady-state conduction but requires at least the time- and depth-dependent heat transfer equations with a damping mechanism. The damping might be storage and release of heat through convection and/or the phase change of water from liquid to vapor and back within the woodchip layer. Overall, relative cover material effectiveness can be ranked in Fig. 8 as most efficient (Fig. 8b), efficient (Fig. 8a), and least efficient (Fig. 8c).

## 4.5 Summer 2019

The 9300 m$^3$ snow pile emplaced in 2019 lost volume at an average rate of 15 m$^3$ d$^{-1}$ (min of 5 m$^3$ d$^{-1}$ in early July and max of 25 m$^3$ d$^{-1}$ between April and May, when the snow pile was compacting and being covered by woodchips). Between the initial TLS survey in March and the last survey in October, the pile lost 2388 m$^3$ CE6 of snow, a 40% volume loss (not including woodchips). The average percentage loss per day was 0.16% of the initial volume. In comparison to the 2018 snow piles, the pile lost volume more uniformly; no crevices formed and no slumping occurred (Fig. 9), although the surface did become rougher by October, and we noted more surface lowering near dark-colored rocks and logs emplaced to hold down the white, reflective covering. Volume loss between 11 May and 25 August (the most intensive melt season) was similar in all four quadrants of the pile, each of which experienced an average of 0.9 m lowering. More lowering occurred on the pile boundaries, specifically along the western margin, as shown clearly by the blue and purple colors in Fig. 9a.

## 5 Discussion

Data we collected allow us to (1) determine the volumetric change rate of small snow piles stored over summer with dif-

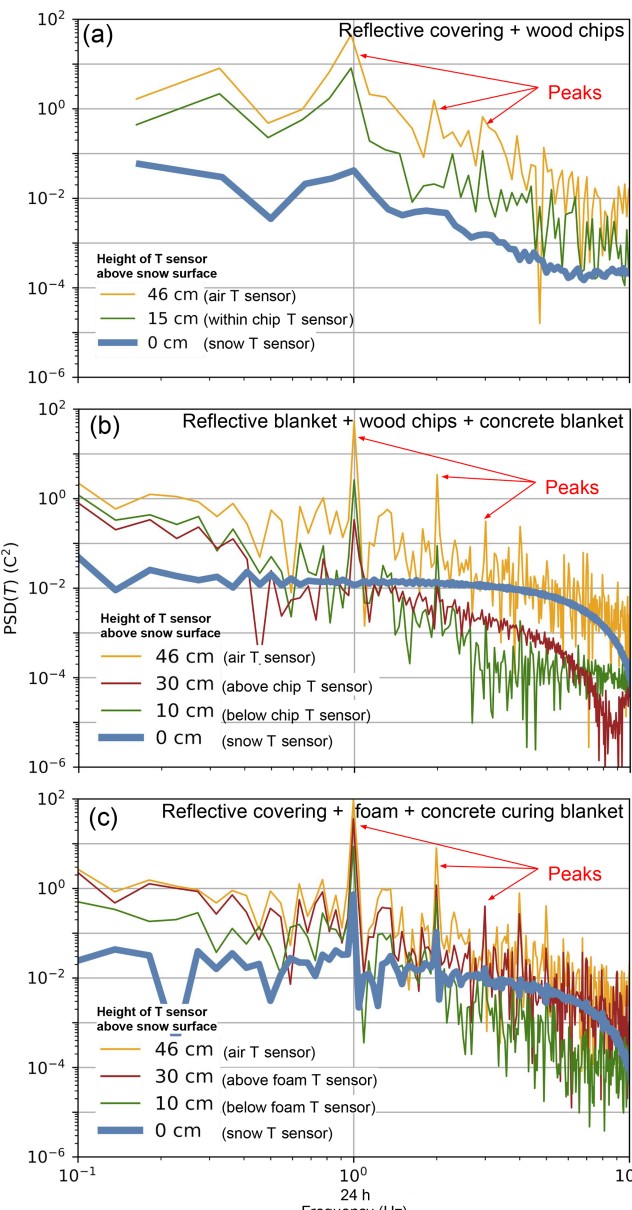

**Figure 8.** Power spectral density of temperature records from three different cover experiments (Fig. 4b, e, and f). PSD normalizes frequency to 24 h = $10^0$ and displays the magnitude of each temperature oscillation frequency for each sensor per experiment (depth in centimeters measured above sensor at the snow – 0 cm). **(a)** Experiment with woodchips and reflective cover (Fig. 4b). **(b)** Experiment with a concrete curing blanket, woodchips, and a reflective cover (Fig. 4e). **(c)** Experiment with concrete curing blanket, open-cell foam, and a reflective cover (Fig. 4f). The lack of detectable signal (flat blue line) at snow level (0 cm) in **(b)** demonstrates that three-layer configuration with woodchips best damps the diurnal temperature signal. Colors correspond to colors from Fig. 4.

ferent coverings, (2) suggest an optimal snow preservation strategy for low-elevation, midlatitude sites based on these data, and (3) test this optimized snow storage strategy at scale.

## 5.1 Experimental snow pile melt rate

The survival of small ($200\,\mathrm{m}^3$) snow piles through the warmer-than-average summer of 2018 and the results of both repeated TLS surveys and continuous in situ thermal data collected during a variety of different snow cover experiments suggest ways of optimizing over-summer snow storage at low elevations and midlatitudes. The 2018 snow piles experienced nonuniform cover and nonideal geometry and developed crevices that exposed snow to direct sunlight, all of which increased the rate of snowmelt and thus volume loss. Field observations and TLS surveys demonstrated that the thickness of woodchips covering the snow was not uniform and became less uniform over time as melt changed the pile shape (Fig. 3). Woodchip depth changed over the summer as crevices, which grew over time, exposed bare snow to direct sunlight, which led to rapid and nonuniform pile melting (Fig. 6). Crevices formed along boundaries of the large plastic sheets, which were emplaced to prevent woodchips from mixing with the snow. Openings in the woodchip cover also resulted from snow slumping within the pile – both piles had steep sides, and the DSMs revealed snow moving downslope (Fig. 6). Lintzén and Knutsson (2018) reference similar snow pile and cover failure due to steep pile-side geometry.

Snow pile size impacts the rate of volumetric change significantly. The two test piles were small, only a few percent of the volume of snow typically stored over summer by Nordic ski areas. For example, in Davos, Switzerland, and Martell, Italy, test piles were about 6000 and 6300 $\mathrm{m}^3$ (Grünewald et al., 2018). The Nordkette nordic ski operation in Innsbruck, Austria, stores $\sim 13\,000\,\mathrm{m}^3$ of snow, and Östersund, Sweden, stores 20 000 to 50 000 $\mathrm{m}^3$ piles. Small piles have a larger surface-area-to-volume ratio (SA / $V$), which allows more effective heat transfer through radiation, conduction, and latent heat transfer. A simple comparison of two hemispheres, one containing $200\,\mathrm{m}^3$ of snow and the other containing $9000\,\mathrm{m}^3$ of snow, indicates that SA / $V$ changes from 0.66 to 0.23 between the smaller and larger pile. As larger piles have a SA / $V$ ratio that is 3 times lower in comparison to smaller piles, there is comparatively less snow near the surface thermal boundary, which decreases heat transfer per unit snow volume and thus the melt rate as a percentage of pile volume.

## 5.2 Optimal approach for over-summer snow preservation at midlatitude and low-elevation sites

The 2018 survival of snow through the summer in small piles with only simple woodchip, foam, and reflective coverings suggested that larger piles, using an optimized cover strategy,

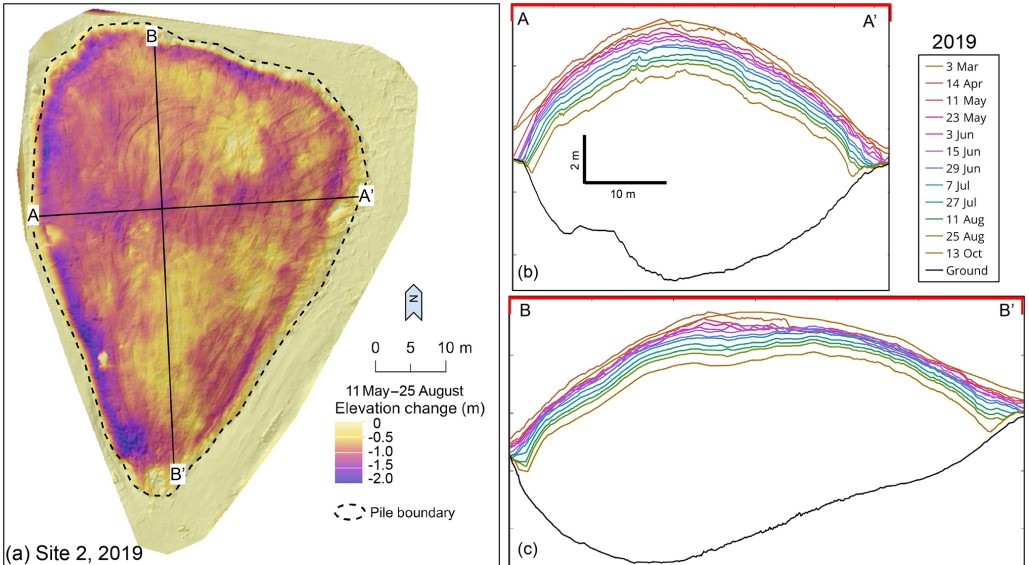

**Figure 9.** Volume change of 2019 snow pile. **(a)** Spatial variability of elevation change 2019 snow pile between 11 May and 25 August 2019. Cross sections $A$–$A'$ and $B$–$B'$ are marked in black. **(b)** Profile for each terrestrial-laser-scan survey (3 March to 13 October 2019; $n = 12$). **(c)** Profile for each terrestrial-laser-scan survey (2 March to 13 October 2019; $n = 12$). Each scan represented by a colored line in panels **(b)** and **(c)**.

will allow for practical over-summer snow storage at midlatitude ($< 45°$ N) and low-elevation ($< 350$ m a.s.l.) locations. Our results are encouraging given the relative warmth of the 2018 summer season, the simple and spatially inconsistent nature of our cover material ($20 \pm 10$ cm of woodchips), and the small size of the test piles ($\sim 200$ m$^3$). Previous snow storage studies found success with woody covers as well but in different geographic settings. Grünewald et al. (2018) suggested that a 40 cm layer of sawdust sufficiently optimized snow retention in Davos, Switzerland, and Martell, Italy. Skogsberg and Nordell (2001) reported that woodchips reduced snowmelt by 20 %–30 % at the Sundsvall Hospital in Sweden. Lintzén and Knutsson (2018) built snowmelt models and ran field tests in northern Scandinavia, revealing that thick layers of woody materials successfully minimized snowmelt. In practice, financial constraints often control the choice of cover strategies. For example, the thicker the layer of woodchips, the better protected the pile will be and the less over-summer melt will occur. However, using more chips increases cost (Grünewald et al., 2018).

The experimental data (Fig. 4) show that the magnitude of daily temperature oscillations at the snow surface below the covering (blue line in all panels) is highly dependent upon the cover strategy. For example, in Fig. 4c, the temperature within the rigid foam board increases above air temperature (purple line increasing above the yellow line). Due to the rigidity of the foam boards and the nonuniform melting of the pile, the foam shifted and exposed snow to direct solar radiation, allowing warm air to move between the snow and the foam. Such failure of the cover system allowed tempera-

tures at the snow interface to rise significantly above 0 °C. The three-layer cover (insulating blanket, wet woodchips, and reflective cover) minimizes heat transfer into the stored snow, as evidenced by the lack of diurnal temperature oscillations at the snow surface during this and only this experiment (Fig. 4e). The comparison between foam and saturated woodchips PSDs (Fig. 8) shows the dramatic effect on the heat transfer from the atmosphere to the snow caused by the high heat capacity and thus thermal inertia of wet woodchips. The damping of diurnal temperature peaks by the three-layer cover system suggests that it will be the most effective for preserving snow over the summer.

Although the relevant heat transfer mechanisms remain uncertain, Fig. 8 demonstrates the effectiveness of the three-layer cover approach to buffering heat transfer from the environment to the snow. Deducing specific heat transfer mechanisms will require different and more complex measurements, as heat transfer is dependent on not only air temperature but also surface temperature, long-wave radiation, and turbulent fluxes. Perhaps evaporation of water from the wet woodchips absorbs thermal energy during the day which is released as the latent heat of condensation at night when the reflective blanket cools – effectively increasing the thermal mass of the woodchip layer. Depending on weather conditions, which influence long-wave radiation through cloudiness and turbulent fluxes through wind, the heat transfer may be directed toward the snow pile (warm nights) or radiated to the atmosphere (cold nights). In any case, the large thermal mass of wet woodchips, in concert with an underlying layer (the concrete curing blanket), and rejection of short-wave in-

cident radiation from sunlight by the reflective cover, appears more important than the insulating capability ($R$ value) of the cover material in damping daily temperature fluctuations at the snow surface.

## 5.3    Summer 2019, testing the optimized snow storage strategy at scale

Field data, TLS, and thermal observations from the 2018 experiments allowed for a full-scale test of our optimized snow storage strategy in 2019. Optimization began by further excavating the storage area so that the resulting pile would sit within a pit and have gently sloping sides to reduce the chance of mass movements and crevassing on the pile margins. Snowmaking was tuned so that the density of the snow emplaced was already high; this minimized settling after covering. The snow was then compacted by repeated passes of large excavators and PistenBully groomers. Letting the snow settle and transform before covering also reduced the chance of mass movements which, in 2018, compromised pile and cover integrity. Results from the 2018 cover material experiments (most effective was a reflective cover, woodchips, and a concrete curing blanket) informed the 2019 covering method (Fig. 8). Rather than use metallized cover material, which was expensive, fragile, and impermeable, we used a high-albedo (0.75), white, permeable geofabric that allowed rain to infiltrate, thus mitigating regulatory concerns related to a large impermeable area. Concrete curing blankets were not used in 2019 due to cost and logistical complications of emplacement.

The 2019 pile, using an optimized strategy, confirmed the viability of snow storage at the COC. The most rapid volume losses in 2019 were in the midsummer; while they were higher in absolute terms than those in 2018 because the pile was 45 times larger, they CE7 were more than 3 times lower in percentage terms. Most melt was focused along the western boundary, perhaps because the snow here was thin or not as thickly covered by woodchips or because western sun exposure occurs late in the day when the air temperatures are warmer; there is likely less net radiative cooling along the western side of the pile, as there is a steep, forested slope immediately adjacent to the snow storage area. Compared with the average percentage loss per day of the 2018 piles (0.64 % per day), the 2019 snow pile average percentage loss per day was 0.16 %. We suspect that the difference in volume loss reflects primarily the surface-area-to-volume ratio of the 2019 snow pile, which is about 3 times less than the small piles tested in 2018. A 3-fold change in the SA / $V$ ratio compared with a 4-fold reduction in the percentage volumetric change rate suggests the impact of an improved cover strategy. The complete covering of the 2019 pile with a reflective geofabric likely slowed melt by rejecting short-wave radiation as well as protecting the snow even if the woodchips shifted. TLS imagery from 2019 demonstrates that gentle side slopes of the pile prevented any large mass movements of snow, indi-

cating that pile shape and snow pre-consolidation are important (Fig. 9).

TLS data show that from April until mid-October, more than 60 % by volume of the snow initially placed in the April 2019 pile remained. Considering the snow density data gathered from the 2018 piles, which increased from 500 to 700 kg m$^{-3}$ over the summer, some of this volume loss could be accounted for by compaction rather than melting. This suggestion is supported by the lack of surface water draining from the 2019 pile, which is underlain by relatively impermeable rock and clay-rich glacial till. With fall temperatures and the sun angle dropping, incident solar radiation as well as convective and conductive heat transfer are diminished greatly from midsummer values. This means that the COC will have > 5000 m$^3$ of snow to spread in November for early-season skiing. Covering 5 m wide trails 50 cm deep will allow at least 2 km of skiing at opening and will provide a base so that any natural snow that does fall will be retained.

## 6    Conclusions

Data presented here show that snow storage at midlatitudes and low elevations is a practical climate change adaptation that can extend the nordic ski season and the sport's viability as the climate continues to warm. Using 14 terrestrial-laser-scan surveys between March and September 2018, we determined rates of volumetric change of two 200 m$^3$ snow piles covered in woodchips. Average volume loss rates were 1.24 and 1.50 m$^3$ d$^{-1}$, with the highest rates of volumetric change in July and the lowest rates of volumetric change in September. A three-layer cover approach was most effective: a concrete curing blanket, a 20 cm layer of woodchips, and a reflective covering. This cover approach reduces solar gain and buffers the effect of > 30 °C summer daytime temperatures and high (> 78 %) relative humidity on stored snow. Using data collected during summer 2018, we tested our experimental results in summer of 2019 by creating a 9300 m$^3$ snow pile. Due to cost and logistical issues, we covered the pile using a two-layer approach – 650 m$^3$ of woodchips and white, permeable geofabric. The average volume loss rate between March and October was 15 m$^3$ d$^{-1}$ (or 0.16 % of the initial volume per day). About 5600 m$^3$ CE8 of snow remained as the melt season ended in mid-October. This quantity of snow is sufficient for the COC to open their 2019 season and represents > 60 % retention of snow by volume, comparable to storage losses at other storage sites (at higher elevation and latitude). Future research could analyze financial and environmental feasibility of snow storage at different global locations and focus on heat transfer mechanisms of different cover materials. Research could also explore other climate change adaptation strategies for nordic ski centers that minimize carbon emissions and maximize operational success.

*Data availability.* Data are available at https://doi.org/10.1594/PANGAEA.899744 (Weiss and Bierman, 2018).

*Author contributions.* PB and YD co-conceptualized the experiment. HW and YD curated the data. HW, YD, and SH conducted the TLS and PSD analysis. HW and PB acquired funding, developed the methodology (assisted by SH), conducted the investigation, and validated data. HW, PB, and YD prepared data visualizations. HW and PB wrote the original paper draft. All authors contributed to the review and editing process.

*Competing interests.* The authors declare that they have no conflict of interest.

*Acknowledgements.* We thank Judy Geer and Dick Dreissigacker, directors of the Craftsbury Outdoor Center, for enthusiastically supporting this project. We also thank our editor, Jürg Schweizer, and reviewers, Thomas Grünewald, Nina Lintzen, and one anonymous reviewer, for providing constructive criticism to strengthen this manuscript. Field work was generously assisted by Landon Williams and Amelia Murtha. We acknowledge that the land the research was preformed upon was once the land of the Abenaki people.

*Financial support.* This research has been financially supported by the University of Vermont's Department of Geology, Department of Mechanical Engineering, Rubenstein School of Environment and Natural Resources, the Office of Undergraduate Research, and the Graduate College. Additional support was provided by the National Science Foundation under CMMI-1229045. CE9

*Review statement.* This paper was edited by Jürg Schweizer and reviewed by Thomas Grünewald, Nina Lintzen, and one anonymous referee.

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

## Remarks from the language copy-editor

## Remarks from the typesetter