# Peer review of "Optimization of over-summer snow storage at mid-latitude and low elevation"

_The Cryosphere, 2019_

## Referee Comment (RC1) · Nina Lintzen (Referee) · 5 May 2019

General comments:

The paper presents over-summer snow storage at mid-latitude and low elevation. The tests were performed in Vermont, USA. The goals of the research (according to the statements in the introduction) was to: 1) Determine the melt rate. 2) Infer the environmental factors that most influence snow melt. 3) Suggest an optimized insulation strategy based on the data. I would have liked to see clear responses to all these questions in the conclusions section.

The climate in Europe is warmer than in North America at a similar latitude. A comparison between actual weather data from other over-summer snow storages with warm

summer climate (for example in Europe, Russia and South Korea) would have been desirable.

Specific comments:

The results should be discussed and explained more in detail. For example, what do we see in Figure 4? How much did the temperature change between the different test methods? The scaling in the figures is not so clear so this is obvious just by looking at the figures. I think the results are very interesting but a detailed comparison of foam with and without reflective cover, how much the temperature changed in the "between-foam-spot" etc. would have given more depth to the study. Similar for figures a and b as well as e and f. How much lower was the temperature above the concrete curing blanket if you compare e and f? In figures 4 c and d, the temperature on the snow seems to be much higher than 0°C in the end of the experiments. Is this due to some measurement error? Or how do you explain this temperature increase?

The PSD and the results in Figure 8 needs to be explained more in detail. What is the PSD? How do you calculate the PSD? What do we actually see in the figures?

I would suggest to enlarge and develop the discussion section. Discuss the three goals with this research and compare them to other studies. Are there for example other studies where the melt rate has been studied and how do your results relate to these? Which were the environmental factors that most influenced the snow melt and how did you reach this conclusion?

  Comments from the text:

Page 6, # 15: How do you conclude that larger piles using an optimized insulation strategy allow for efficient over-summer snow storage from these experiments? For sure this is possible, it has been done at places with warm climate (for example in Sochi, Russia and Pyeongchang, South Korea).

Page 6, # 30: The planned snow storage for the summer 2019 is interesting, but not

relevant for this presented study and experiment.

Page 7, # 20: Conclude answers to your three research questions. Also, conclude and point out that based on your experiments and from the different experimental setups you tested, the three layer insulation was the best. Scaling up from 200 m3 to 7000 m3 will increase the remaining amount of snow, but this is not a conclusion from the performed tests in this study. Scaling up to any larger volume will render a larger remaining volume of snow, but this is not a relevant conclusion from the tests performed in this presented study. However, in the discussion section I would suggest that you mention the fact that larger volumes of snow will increase the efficiency of snow storage, as have been seen in previous studies, and as you have mentioned in #25 and 30 on page 6.

Technical comments:

Page 4, # 5: "man-made" snow should be changed to "machine-made snow".

Page 4, # 35: Were the sheets of plastic and wood chips removed from the whole pile or just from the 1 m2 test area?

Page 5, # 5: It says that the humidity remained high, but how high is a high humidity? A number would have been interesting.

---

## Referee Comment (RC2) · Anonymous Referee #2 · 6 May 2019

The paper, which summarizes a small field test of snow storage in Vermont, US, is concisely and nicely written. Different types of thermal insulation were tested and evaluated by careful measurements of temperature and melting at two test plants, which both contained 200 m3 snow. Similar tests have previously been carried out in e.g. Sweden, Austria and Japan. Still I think it is a good paper since it suggests the most appropriate thermal insulation for a climate where such snow storage technology would be applicable. This paper has valuable information for future snow storage projects and should be accepted for publication in The Cryosphere.

Minor comments:

SI units are generally used. There are some examples where it is not (80 cal g-1) (g m-3) and these could be changed.

[Figure]

Page 1, Line 31 says: Earth's climate is warming in response to the addition of CO2 and other greenhouse gasses to the atmosphere (Steffen et. al.,2018). Suggested revision: Earth's climate is warming (Steffen et. al.,2018).

Page 5, section 1; The maximum rate of precipitation is given. I miss the mean annual precipitation, which should be added..

The paper refers to research done in Sweden, Austria and Japan but there is no reference to Ed Morofsky, who was involved in ice and snow storage research in Canada? I also miss that they did not do any calculations of the heat transfer through the thermal insulation.
* * *

---

## Referee Comment (RC3) · Thomas Grünewald (Referee) · 7 May 2019

Weiss et al present a case study on over-summer snow storage (snow farming) at two sites in Vermont, US. Melt rates of two small snow piles were calculated from repeated high resolution snow volumes measured with terrestrial laser scanning (TLS). Meteorological parameters and temperatures in the covering layer were continuously measured. Moreover and they investigate the performance of different settings of covering materials (combination of wood chips, open-cell foam, rigid foam, blanket); It is shown that snow storage seems possible, even at such a low-elevation site. The novelty of the study is the high temporal resolution of the snow volume surveys (14 surveys over summer-season) and the detailed assessment of temperature gradients within the covering-material. Such data have not been presented before. Data and

results are generally presented nicely and are definitively worth publication in TC after a careful revision; some sections are unclear and need to be reformulated or enlarged (see below). Most important, I think that the large potential of the data set is not fully exploited: The high spatial (10cm) and temporal (about 2 weeks) resolution of the TLS data would allow a more detailed analysis (see specific comments). Considering the effort of the suggested additional analysis and the many smaller things to be changed I suggest major revision (could also be major minor revision);

Specific comments:

1) TLS section requires more detailed information (settings of device, accuracy, references)

2) Section 5.1 should be enlarged with an analysis on spatial and temporal variability of snow melt (TLS data).

Interesting questions to be answered are: How do melt rates principally vary spatially (e.g. depending on slope and aspect of the piles)? How does the type of covering material combination affect melt rates? (Compare the different areas) How does the spatially varying depth of the wood chips (known from first survey) affect melt? Addressing these questions would be very interesting and would substantially improve the impact of the paper.

3) Section 5.2 must be revised; Temperature alone cannot be used as criterion to judge covering material performance; TLS data could be used to analyze effects of different cover on snow melt;

4) Results should be related to earlier studies and other snow farming projects;

5) Many statements need to be rephrased for correctness and more clarity

More details can be found in the technical comments below.

Technical comments:

[Figure]

Abstract: should be a single paragraph. Remove line-breaks

p1 l 13: this statement "has never been attempted at low elevations..." is too rigid. There are some low-elevated places (e.g. Ruhpolding Germany, elevation 700m) that successfully operated snow farming for many years. Please formulate more carefully.

L 22-24: It is unclear how the two piles were covered and to which pile the mentioned rates of change refer; what is meant with "minimum rates of change"? I suggest to provide ranges and mean for the rates of change.

L25: replace "blackbody radiation" with "long-wave emission"

L32-33: "This warming... snow packs." This statement requires a reference

L36 in that context it is unclear what is meant with "... by covering snow". Please reformulate; moreover the current review paper of Steiger et al 2017 could be cited in that context;

P2 L1-3: there was only little research on snow making (from the science side) in the last decades; most of the innovation came directly from industry; This changed a bit in the last years when the public sector and science began to realize the importance of snow making and snow management and the challenges of climate change for the skiing industry; Examples for recent publications are Hanzer et al. 2014, Grünewald and Wolfsperger 2019 or Spandre et al. 2016;

L 6: why is snow storage safer than relying on weather conditions? Please be more concrete here

L8-14: For cooling people mainly used lake or river-ice; the cited reference (Nanegast 1990) also seems to refer to ice; snow was (and is still used) in some areas of Asia and Scandinavia. As formulated now, the paragraph is bit confusing; Please reformulate and be careful not to mix ice storage with snow-farming for winter sports as described in the end of the paragraph;
L14: snow storage is quite expensive (see Grünewald et al. 2018)

L16 Besides solar radiation, air temperature is most important for snow melt (see Fig 11 in Grünewald et al. 2018); precipitation is less relevant; why should evaporative cooling be higher in cold and dry climates? Evaporation is depending on the temperature gradient between surface and air, wind and wetness of the covering-layer.

L22 I suggest to point out the research gap and the novelty of the study here

L27 use J/kg as unit for energy instead of cal/g

L31 use long wave emission or long wave radiation instead of blackbody radiation

L34 Long wave radiation especially depends on surface temperature (Stefan Blozmann law: power of 4!)

L36 snow melt instead of snowpack melt

P3 L5 I am not happy about the formulation "high elevation"; if 1600 is high, what is 3000m? And: the latitude of Vermont (45°) was called "low" (P2 L17);" here a very similar latitude of 46° is called "mid"; this is not consistent;

L6 I suggest to write machine-made or technical snow instead of artificial snow

L6 remove "wet"

L8 write "Using a physically based model" instead of "thermal models"

L8/9 please clarify context: most effective means in relation to work/cost effort; deeper layers can safe more snow but the effort is higher

L11 write "capillary flow" instead of "capillary action"

Section 3: The section is very short. I suggest to merge section 3 and 4 to "Methods and settings" and then to introduce subsections; (e.g. study site, Weather stations, terrestrial laser scanning, snow density, insulation experiments...)

L30 what is the elevation of the site?

L33 What is the elevation of the station?

L 31-33: please also indicate mean temperatures not only minimum and maximum

L 34-36 USAD, NOAA, USGS > citation style is wrong; year is missing

P4 L1 please describe differences between the two sites (pile 1 and 2), e.g. shadow, slope . . ..

L5 provide a reference to snow density section

L5 provide more information on the properties of the plastic sheets (e.g. thickness, size, water permeability, thermal conductivity . . .) and for what reason they were used (I guess to reduce snow pollution as stated later); such information should also be given for the foam used in for the insulation experiments

L6 brackets are missing (Fig. 3)

L9 at which height above ground were the meteorological measurements performed?

L12 be more clear about soil temperatures: how many sensors? Where were the sensors? Where the sensors in the ground or in the covering layer?

L15ff this section requires more details: the dates of the scans should be provided, e.g. in a table; Also add a table with the technical specifications of the laser scanner; Was multi-station adjustment used for registration; why not? It is an easy approach to improve registration of the data; What is the accuracy of the data? Were data gaps (scan shadows) existing? How were they handled? If a direct accuracy evaluation of the data is not possible, at least references to earlier studies that assessed TLS accuracy in similar settings should be added, e.g. Prokop et al. 2008, Grünewald et al. 2010, Grünewald and Wolfsperger 2019;

L32 please add for how long the insulation experiments lasted; until end of summer?

L32 please state what kind of R (e.g. Pearson's correlation coefficient) is used

P5 Sect 5.1 Sum of precipitation should also be given; How were condition of the recorded summer season in relation to long term climate? Data from station COC described in Sect. 3 could be used to rate this summer;

L10-17: It is not clear which measurements are described here: the sensor below the piles or the ones next to the piles? Is there an explanation for the much larger T – variability for the 5 cm sensor at site 1 in relation to site 2? To which of the two sites does Fig 5 refer to?

L17-19 unclear: only measurements of one site (below pile or next to pile) are shown in Fig 5;

L24: add a reference to Fig 3 (after ..."for pile 2.")

L25 use kg/m3 instead of g/m-3

L25-26 Where were densities measured (in which depth) obtained? Densification should be related and discussed in relation to the results of Grünewald et al. 2018 who showed an increase in density, both in time and in depth;

L26-27 "Relative to ... (0.9g/cm-3)." Relating density to fresh snow is not meaningful in that context and could be removed;

L27: I do not think that this is an adequate explanation. Snow with a density of 500 kg/m3 should already be fully decomposed and rounded; Was the snow dry during density measurements? Or was there some liquid water content? Or did you identify ice aggregations resulting from refrozen water? What was the grain size in March?

L29: Please check the numbers: Considering the very similar melt rates of the two sites (Fig 7) the difference between 1.24 and 1.5 m3/d seems very high; is the removal of the 30m3 snow possible part of the melt rate?

L29-32 Discussing melt rates is the main focus of the paper; Please discuss them in

more detail; Your data set should allow a much more detailed analysis! e.g. how do melt rates change in time and how does this related to meteorology? Do melt rates vary spatially? What is the difference between the two piles? What is the difference between sections with different cover material?

L32-37 possibly even the effect of the crevasses could be seen in the TLS data (e.g. local changes in melt rates?

P6 Sect 5.2. This section is pretty poor. It should be enhanced: a discussion and reasoning on the effects of the different covering types (properties of materials and how do they interact with snow and atmosphere is missing; Currently only temperatures are analyzed but this is not enough to judge performance of the different materials; The TLS data could be used to quantify and discuss if and how volume losses differ under different covering materials.

add references to the specific panels for Fig 4

L2 insulation efficiency is not only a function of T, e.g longwave emission or turbulent fluxes are not only depending to T but very relevant for the energy balance;

L12 the presented experiments used wood chips and a plastic planked not only wood chips;

L13-14 climate is not only a function of latitude and elevation; please rephrase

L 15 (fairbanksmuseum, 2019)) > remove bracket

L36 Provide more details on the PSD method; how does it work and what is its benefit? How is it interpreted? Add references;

P7 L6-9 This explanation is too simple: heat transfer is not simply depending on air temperature; surface temperature, cloudiness (longwave radiation) and wind (turbulent fluxes) are also crucial; See discussion of simulation results in Grünewald et al. 2018 and the sections about energy balance, and snow melt of the recent review paper of

Mott et al. 2018; these references and possible also other earlier work should be cited in context of the discussion;

L11 what is the "R-value"?

Section 7: Conclusions should be prolonged; Here all three research questions form the introduction should be shortly answered; an outlook on future research that might be useful to enhance our understanding on snow storage might also be added;

Figures

Figure 1 b) it would be nice if the list would be ordered geographically; Several sites are missing (see attached pdf; Reference: Wolfsperger et al 2018)

Figure 4: T fluctuation of the blue line is hardly visible; possibly change axis or figure dimension

Figure 5: Figure a should be enlarged vertically to improve readability; grids or vertical lines should be added; For humidity and radiation adding daily mean values as line could also help to improve readability; Legend: To which snow pile does the figure refer to? Ground temperatures below or next to pile?

Figure 6: Please add a legend relating colors to dates.

Figure 7: Why is the increase in volume from April 1 to May 1 for site 2 so much larger than for site 1? Was there such a big difference in volume of chips added? Are colors between the two panels possibly mixed? The huge melt rate drop on July 1 might be correct for Site 1 (blue) but not for site 2; Add a grid or horizontal lines for readability;

P9 L15 doi seems to be wrong

L22 and L 24 The papers are not cited in the text;

Having only checked few selected references I found three mistakes; I guess that there are more. Please check your citations and references carefully!

Best regards Thomas Grünewald

Mentioned Literature

Steiger, R., Scott, D., Abegg, B., Pons, M., and Aall, C. (2017). A critical review of climate change risk for ski tourism. Curr. Issues Tour. 1–37. doi: 10.1080/13683500.2017.1410110

Hanzer, F., Marke, T., and Strasser, U. (2014). Distributed, explicit modeling of technical snow production for a ski area in the Schladming region (Austrian Alps). Cold Reg. Sci. Technol. 108, 113–124. doi: 10.1016/j.coldregions.2014. 08.003

Spandre, P., Morin, S., Lafaysse, M., Lejeune, Y., François, H., and GeorgeMarcelpoil, E. (2016). Integration of snow management processes into a detailed snowpack model.Cold Regions Sci.Technol.125,48–64.doi:10.1016/j. coldregions.2016.01.002

Grünewald T and Wolfsperger F (2019) Water Losses During Technical Snow Production: Results From Field Experiments. Front. Earth Sci. 7:78. doi: 10.3389/feart.2019.00078

Prokop, A., Schirmer, M., Rub, M., Lehning, M., and Stocker, M. (2008). Acomparison of measurement methods: terrestrial laserscanning, tachymetry and snowprobing, for the determination of spatial snowdepth distributionon slopes.Ann.Glaciol.49,210–216.doi:10.3189/172756408787814726

Grünewald, T., Schirmer, M., Mott, R., and Lehning, M. (2010). Spatial and temporal variability of snow depth and ablation rates in a small mountain catchment.Cryosphere4,215–225. doi:10.5194/tc-4-215-2010

Wolfsperger, F., Rhyner, H. U., and Schneebeli, M. (2018). Pistenpräparation und Pistenpflege. Das Handbuch für den Praktiker. Davos, CH: SL-Insitut für Schnee- und Lawinenforschung SLF.

Grünewald, T., Wolfsperger, F., and Lehning, M. (2018). Snow farming: conserving

snow over the summer season. Cryosphere 12, 385–400. doi: 10.5194/tc-12-385-2018

Mott R, Vionnet V and Grünewald T (2018). The Seasonal Snow Cover Dynamics: Review on Wind-Driven Coupling Processes. Front. Earth Sci. 6:197. doi: 10.3389/feart.2018.00197

Please also note the supplement to this comment:
https://www.the-cryosphere-discuss.net/tc-2019-56/tc-2019-56-RC3-supplement.pdf

**Supplement:**

Tab. 7.1: Übersicht über bekannte Snowfarming-Projekte[8].

| Ort | | | Abdeckung | Ertrag | |
|---|---|---|---|---|---|
| Land | Gemeinde | m ü. M. | Material | Volumen [m³] | Verlust [%] |
| AT | St. Jakob im Walde | 1150 | Sägespäne | 2700 | 26 |
| AT | Ramsau | 1100 | Hackschnitzel / LKW-Plane | 20000 | 40 |
| AT | St. Gallenkirch (Montafon) | 2080 | Folie | 15000 | 80 |
| AT | Seefeld (Tirol) | 1200 | Hackschnitzel | 5000 | 40 |
| AT | Hochfilzen | 960 | Vlies | 8000 | k. A. |
| AT | Reiteralm | 2100 | Stroh | k. A. | k. A. |
| AT | Saalbach-Hinterglemm | 1000 | Hackschnitzel / Vlies / Folie | 15000 | 17 |
| AT | Hermagor | 1200 | Stroh | 400 | 25 |
| AT | Kitzbühel | 1900 | Isolierplatten / Silofolie / Vlies | 25000 | 20 |
| CAN | Canmore Nordic Centre | 1380 | Sägespäne | k. A. | k. A. |
| CH | Davos (altes Depot) | 1650 | Sägespäne | 6900 | 22 |
| CH | Davos | 1650 | Sägespäne | 16000 | 16 |
| CH | Disentis | 2600 | Vlies | k. A. | k. A. |
| CH | Davos Jakobshorn | 2600 | Vlies | 20500 | 57 |
| CH | Engelberg | 1050 | Hackschnitzel | 600 | k. A. |
| D | Ruhpolding | 700 | Isolierplatten / Silofolie / Tape | 10850 | 30 |
| D | Neustadt | 820 | Isolierplatten / Folie | 10000 | 20 |
| D | Klingenthal | 569 | Sägespäne | 16000 | k. A. |
| D | Oberhof | 815 | Isolierplatten / Folie | 10000 | k. A. |
| FIN | Ruka | 400 | Sägespäne / Vlies | 30000* | k. A. |
| FIN | Vuokatti | etwa 100 | Sägespäne | 20000 | 20 |
| IT | Livigno | 1800 | Sägespäne / Vlies | 70000 | 25 |
| IT | Corvara | 1900 | Steinbach S500T-550 | 6000 | 50 |
| IT | Martell | 1700 | Hackschnitzel | 7140 | 33 |
| IT | Watles | 2300 | Isolierplatten / Silofolie / Vlies | 25000 | 20 |
| NOR | Söderhamn | etwa 100 | Rinde | k. A. | k. A. |
| NOR | Dovre | 700 | Hackschnitzel | 10000 | k. A. |
| NOR | Beitostølen | 820 | Sägespäne | 18000 | 22 |
| NOR | Trondheim | 180 | Isolierplatten | 18000 | 22 |
| NOR | Geilo | etwa 1000 | Vlies / Sägespäne oder Stroh | k. A. | k. A. |
| RUS | Rosa Khutor | 1600 | Vlies / Isoliermatte | 800000* | 20-50 |
| SE | Östersund | 372 | Sägespäne | 55200* | 24 |
| SE | Orsa | etwa 100 | Rinde | 5000 | k. A. |
| SE | Piteå | etwa 100 | Rinde / Vlies | 3400 | 30 |
| SE | Arjeplog | etwa 100 | Rinde / Vlies | 1600 | 60 |

* Gesamtvolumen in mehrere Haufen aufgeteilt
* * *
[8]  Gletscherabdeckungen sind hier nicht aufgeführt. Sie werden jedoch im Alpenraum intensiv eingesetzt (z. B. Kaunertal, Pitztal, Sölden, Diavolezza, Gemsstock, Saas Fee, Vorabgletscher usw.).

---

## Author Comment (AC1) · 9 Jul 2019

Weiss et al. Anonymous Referee #2

Author responses are located below referee comments.

The paper, which summarizes a small field test of snow storage in Vermont, US, is concisely and nicely written. Different types of thermal insulation were tested and evaluated by careful measurements of temperature and melting at two test plants, which both contained 200 m3 snow. Similar tests have previously been carried out in e.g. Sweden, Austria and Japan. Still I think it is a good paper since it suggests the most

[Figure]

appropriate thermal insulation for a climate where such snow storage technology would be applicable. This paper has valuable information for future snow storage projects and should be accepted for publication in The Cryosphere.

Minor comments: SI units are generally used. There are some examples where it is not (80 cal g-1) (g m-3) and these could be changed. Author Response: Thank you for catching this inconsistency and non-SI units will be changed in the revised manuscript.

Page 1, Line 31 says: Earth's climate is warming in response to the addition of CO2 and other greenhouse gasses to the atmosphere (Steffen et. al.,2018). Suggested revision: Earth's climate is warming (Steffen et. al.,2018). Author Response: Thank you for this concision suggestion and the suggested change will be made.

Page 5, section 1; The maximum rate of precipitation is given. I miss the mean annual precipitation, which should be added. Author Response: Thank you for this suggestion. We will include mean to provide additional context.

The paper refers to research done in Sweden, Austria and Japan but there is no reference to Ed Morofsky, who was involved in ice and snow storage research in Canada? Author Response: Thank you for the researcher suggestion – we will cite Morofsky in the revised manuscript.

I also miss that they did not do any calculations of the heat transfer through the thermal insulation. Author Response: Thank you for this note – we did not include heat transfer calculations as the models were not ready at the time of publication, and the inhomogeneous wood chip layer and melt pattern rendered calculations less valuable. We hope to include heat transfer calculations in future publications when we have more consistent data.

Please also note the supplement to this comment:
https://www.the-cryosphere-discuss.net/tc-2019-56/tc-2019-56-AC1-supplement.pdf

---

## Author Comment (AC2) · 9 Jul 2019

Weiss et al. Nina Lintzen (Referee) nina.lintzen@ltu.se

Author responses are below referee comments.

General comments: 1) The paper presents over-summer snow storage at mid-latitude and low elevation. The tests were performed in Vermont, USA. The goals of the research (according to the statements in the introduction) was to: 1) Determine the melt rate. 2) Infer the environmental factors that most influence snow melt. 3) Suggest an optimized insulation strategy based on the data. I would have liked to see clear responses to all these questions in the conclusions section. Author Response: Thank you for this comment – we realized we've focused on 1 and 3 (though not explicitly) but did not address goal 2 in our conclusion. We will make this change in our revision. The data collected allows us to address goals 1 and 3, yet not goal 2 which we will remove from the introduction. We will then more clearly address 1 and 3 in the conclusion.

2) The climate in Europe is warmer than in North America at a similar latitude. A comparison between actual weather data from other over-summer snow storages with warm summer climate (for example in Europe, Russia and South Korea) would have been desirable. Author Response: Great suggestion – we discussed this comparison yet it did not end up in the final paper. We will incorporate this into revision.

Specific comments: 3) The results should be discussed and explained more in detail. For example, what do we see in Figure 4? Author Response: We appreciate this comment – initially, we had extensive narration and decided to simplify the section but perhaps removed too much. We will add more narration in revision.

4) How much did the temperature change between the different test methods? The scaling in the figures is not so clear so this is obvious just by looking at the figures. I think the results are very interesting but a detailed comparison of foam with and without reflective cover, how much the temperature changed in the "between- foam-spot" etc. would have given more depth to the study. Similar for figures a and b as well as e and f. How much lower was the temperature above the concrete curing blanket if you compare e and f? Author Response: Thank you for this analysis of Figure 4. Power-Density Spectrum Analysis (PDS in Figure 5) is more useful for analyzing effectiveness of different insulation test methods than temperature change alone. However, within Fig. 4 we will include ranges for the sensors at the snow-insulation interface to demonstrate temperature differences between insulation types.

Our goal in using PSD is to determine which temperature signals still displayed the diurnal oscillations – if certain insulation combinations damp the temperature signals

more thoroughly than others, these insulations were more effective at preventing heat from radiating into the pile. The individual temperatures were not as important as their signals throughout the week. It's clear that we did not explain PSD in an accessible way and will revise this.

5) In figures 4 c and d, the temperature on the snow seems to be much higher than 0åŮęC in the end of the experiments. Is this due to some measurement error? Or how do you explain this temperature increase? Author Response: Thanks for this note. Due to the rigidity of the foam boards and the non-uniform melting of the pile, the foam shifted and exposed snow to direct solar radiation, as well as allowed warm air to be trapped between the snow and the foam. In panels c and d, we see this reflected in the temperature sensors at the snow interface reading significantly higher values than 0åŮęC. We will make this clearer as it could help the reader understand the ineffectiveness of the rigid foam panels.

6) The PSD and the results in Figure 8 needs to be explained more in detail. What is the PSD? How do you calculate the PSD? What do we actually see in the figures? Author Response: We thank you for identifying the lack of clarity about PSD. We realize in retrospect that we did not explain the concept in an accessible way and will in further revisions.

7) I would suggest to enlarge and develop the discussion section. Discuss the three goals with this research and compare them to other studies. Are there for example other studies where the melt rate has been studied and how do your results relate to these? Which were the environmental factors that most influenced the snow melt and how did you reach this conclusion? Author Response: This restructuring suggestion is very helpful for streamlining our discussion section. There are few studies thus far that address melt rate of snow within the context of snow storage, however none measured at the weekly time intervals at which we measured snow melt. We can infer most influential environmental factors through looking at which insulation combination was best. We'll restructure to address the three goals.

Comments from the text: 8) Page 6, # 15: How do you conclude that larger piles using an optimized insulation strategy allow for efficient over-summer snow storage from these experiments? For sure this is possible, it has been done at places with warm climate (for example in Sochi, Russia and Pyeongchang, South Korea). Author Response: Thanks for the comment. Larger piles have lower surface area/volume ratio of large piles in comparison to smaller piles. We will do a better job of incorporating the SA/V ratio into this section.

9) Page 6, # 30: The planned snow storage for the summer 2019 is interesting, but not relevant for this presented study and experiment. Author Response: Thank you for this observation – if we are short on space, we will remove it. If we do not remove it, we will be sure to more accurately label as "Future Work" to clearly identify it is not part of the current study.

10) Page 7, # 20: Conclude answers to your three research questions. Also, conclude and point out that based on your experiments and from the different experimental set-ups you tested, the three layer insulation was the best. Scaling up from 200 m3 to 7000 m3 will increase the remaining amount of snow, but this is not a conclusion from the performed tests in this study. Scaling up to any larger volume will render a larger remaining volume of snow, but this is not a relevant conclusion from the tests performed in this presented study. However, in the discussion section I would suggest that you mention the fact that larger volumes of snow will increase the efficiency of snow storage, as have been seen in previous studies, and as you have mentioned in #25 and 30 on page 6. Author Response: Thank you for the clarifying and structuring suggestions – you're correct that we did not test the effects of snow melt for different size piles and we will be sure to more clearly define our conclusions based on the insulation experiments alone. Great suggestion to include the larger volume, more snow scenario in the discussion section and could reference this in the conclusion while still staying true to the limitations of our experiments.

Technical comments: 11) Page 4, # 5: "man-made" snow should be changed to

"machine-made snow". Author Response: Thank you - we will change this phrase to remove the outdated gender bias.

12) Page 4, # 35: Were the sheets of plastic and wood chips removed from the whole pile or just from the 1 m2 test area? Author Response: The sheets of plastic and wood chips were removed from just the test areas – we will clarify this in the revision.

13) Page 5, # 5: It says that the humidity remained high, but how high is a high humidity? A number would have been interesting. Author Response: Agreed – we will make this comparison in the next revision.

Please also note the supplement to this comment:
https://www.the-cryosphere-discuss.net/tc-2019-56/tc-2019-56-AC2-supplement.pdf

---

## Author Comment (AC3) · 9 Jul 2019

Weiss et al. Thomas GruÌĹnewald (Referee) t.gruenewald@gmx.ch

Author responses are below referee comments.

Weiss et al present a case study on over-summer snow storage (snow farming) at two sites in Vermont, US. Melt rates of two small snow piles were calculated from repeated high resolution snow volumes measured with terrestrial laser scanning (TLS). Meteorological parameters and temperatures in the covering layer were continuously measured. Moreover and they investigate the performance of different settings of covering materials (combination of wood chips, open-cell foam, rigid foam, blanket); It is shown that snow storage seems possible, even at such a low-elevation site. The novelty of the study is the high temporal resolution of the snow volume surveys (14 surveys over summer-season) and the detailed assessment of temperature gradients within the covering-material. Such data have not been presented before. Data and results are generally presented nicely and are definitively worth publication in TC after a careful revision; some sections are unclear and need to be reformulated or enlarged (see below). Most important, I think that the large potential of the data set is not fully exploited: The high spatial (10cm) and temporal (about 2 weeks) resolution of the TLS data would allow a more detailed analysis (see specific comments). Considering the effort of the suggested additional analysis and the many smaller things to be changed I suggest major revision (could also be major minor revision);

Author response: Thank you for your detailed, constructive comments; they strengthen the manuscript significantly.

Specific comments: 1) TLS section requires more detailed information (settings of device, accuracy, references) Author response: We will revise the manuscript to provide further description of hardware and software settings, registration workflow, and add additional necessary references.

2) Section 5.1 should be enlarged with an analysis on spatial and temporal variability of snow melt (TLS data). Interesting questions to be answered are: How do melt rates principally vary spatially (e.g. depending on slope and aspect of the piles)? How does the type of covering material combination affect melt rates? (Compare the different areas) How does the spatially varying depth of the wood chips (known from first survey) affect melt? Addressing these questions would be very interesting and would substantially improve the impact of the paper. Author response: We appreciate the suggestion for further spatial and temporal analysis of melt from the TLS scans. We will analyze spatial and temporal variability of the melt of the piles and identify if that analysis can help to address the suggested questions. Likely, a spatial/temporal analysis of melt

rates will be inconclusive in addressing some of the questions given the small size of our snow piles and we observed shifting (sliding) of the wood chip insulation over the study period.

3) Section 5.2 must be revised; Temperature alone cannot be used as criterion to judge covering material performance; TLS data could be used to analyze effects of different cover on snow melt; Author response: Our experimental design used temperature as a means of determining insulation efficiency. This makes sense because a change in temperature is directly related to heat flux and melt rate. We only ran insulation experiments on 1m X 1m plots instead of the full pile which means that we cannot spatially nor temporally compare the temperature data from the experiments (collected over 1 week) to the full-pile melt data which was usually acquired every 10-14 days. The Power Density Spectrum (PDS) analysis was included to determine the effectiveness of insulation combinations based on temperature; we will revise these sections to be clearer about both the scope of the insulation experiments and the results from the PDS analysis.

4) Results should be related to earlier studies and other snow farming projects; Author response: We will more explicitly compare this project to other projects within the results and discussion section as it will strengthen the conclusions.

5) Many statements need to be rephrased for correctness and more clarity. More details can be found in the technical comments below. Author response: Thank you for the technical comments; we will address all of them.

Technical comments:

6) Abstract: should be a single paragraph. Remove line-breaks Author response: The change will be made.

7) p1 l 13: this statement "has never been attempted at low elevations..." is too rigid. There are some low-elevated places (e.g. Ruhpolding Germany, elevation 700m) that

successfully operated snow farming for many years. Please formulate more carefully. Author response: Thanks for pointing this generalization out – we meant to indicate the uniqueness of our study site in terms of its combined elevation (300 m asl) and latitude (44°) and will edit this sentence to better reflect our intentions.

8) L 22-24: It is unclear how the two piles were covered and to which pile the mentioned rates of change refer; what is meant with "minimum rates of change"? I suggest to provide ranges and mean for the rates of change. Author response: We will edit to clarify. We will include ranges of snow melt rates.

9) L25: replace "blackbody radiation" with "long-wave emission" Author response: Thanks for the clarification. We will change the phrase.

10) L32-33: "This warming... snow packs." This statement requires a reference Author response: Reference will be added.

11) L36 in that context it is unclear what is meant with "... by covering snow". Please reformulate; moreover the current review paper of Steiger et al 2017 could be cited in that context; Author response: We will add "...by covering snow in various insulative materials to impede snowmelt ". Thank you for the addition of the relevant and recent Steiger paper we will included it in our revision.

12) P2 L1-3: there was only little research on snow making (from the science side) in the last decades; most of the innovation came directly from industry; This changed a bit in the last years when the public sector and science began to realize the importance of snow making and snow management and the challenges of climate change for the skiing industry; Examples for recent publications are Hanzer et al. 2014, GruÌĹnewald and Wolfsperger 2019 or Spandre et al. 2016; Author response: We will amend to put more emphasis on industry's role in the innovation of snow making. Thank you for the paper suggestions.

13) L 6: why is snow storage safer than relying on weather conditions? Please be
more concrete here Author response: Snow storage is safer than relying on weather conditions because optimal snow making conditions are becoming increasingly rare as climate change affects winters. We will revise to more explicitly state this important piece.

14) L8-14: For cooling people mainly used lake or river-ice; the cited reference (Nanegast 1990) also seems to refer to ice; snow was (and is still used) in some areas of Asia and Scandinavia. As formulated now, the paragraph is bit confusing; Please reformulate and be careful not to mix ice storage with snow-farming for winter sports as described in the end of the paragraph; Author response: We will change the statement to be clearer that we're referencing ice storage to demonstrate that organic materials (sawdust/wood chips) have been used in the past to keep vestiges of winter cold – not that we are attempting to compare snow-farming for winter sports to ice houses as these have very different intentions.

15) L14: snow storage is quite expensive (see GruÌĹnewald et al. 2018) Author response: We agree that the insulation process is expensive and we will be clearer that we mean inexpensive compared to a center not being able to open their season on time (and thus, loosing significant business). However, the Cost-Benefit Analysis has not yet been completed so we will be careful with how we discuss this idea.

16) L16 Besides solar radiation, air temperature is most important for snow melt (see Fig 11 in GruÌĹnewald et al. 2018); precipitation is less relevant; why should evaporative cooling be higher in cold and dry climates? Evaporation is depending on the temperature gradient between surface and air, wind and wetness of the covering-layer. Author response: Thanks for pointing this out – high summer relative humidity limits evaporation in Vermont. We will clarify.

17) L22 I suggest to point out the research gap and the novelty of the study here Author response: Thank you for the suggestion – we will include the research gap and novelty when discussing the goals of the research to put it into context.

18) L27 use J/kg as unit for energy instead of cal/g Author response: We will make this edit.

19) L31 use long wave emission or long wave radiation instead of blackbody radiation Author response: We will make this edit.

20) L34 Long wave radiation especially depends on surface temperature (Stefan Blozmann law: power of 4!) Author response: We will include surface temperature when discussing what affects longwave radiation.

21) L36 snow melt instead of snowpack melt Author response: We will make this edit.

22) P3 L5 I am not happy about the formulation "high elevation"; if 1600 is high, what is 3000m? And: the latitude of Vermont (45âŮẹ) was called "low" (P2 L17);" here a very similar latitude of 46âŮẹ is called "mid"; this is not consistent; Author response: We will revise the manuscript for consistency.

23) L6 I suggest to write machine-made or technical snow instead of artificial snow Author response: We will change "artificial" to "machine.

24) L6 remove "wet" Author response: We included the fact that the wood chips contained moisture because moisture plays a key role in reducing snowmelt; dry wood chips would not have been as effective at preserving snow.

25) L8 write "Using a physically based model" instead of "thermal models" Author response: We will make this change.

26) L8/9 please clarify context: most effective means in relation to work/cost effort; deeper layers can safe more snow but the effort is higher Author response: It is important to define "effective" in this context and we will make this change.

27) L11 write "capillary flow" instead of "capillary action" Author response: We will make this edit.

28) Section 3: The section is very short. I suggest to merge section 3 and 4 to "Methods

and settings" and then to introduce subsections; (e.g. study site, Weather stations, terrestrial laser scanning, snow density, insulation experiments...) Author response: We will make this edit.

29) L30 what is the elevation of the site? Author response: We will include the elevation (∼360 m asl).

30) L33 What is the elevation of the station? Author response: We will include the elevation of the weather station (∼215 m asl).

31) L 31-33: please also indicate mean temperatures not only minimum and maximum Author response: We included only min and max because these represented both best and worst case scenarios for summer weather. We will include mean.

32) L 34-36 USAD, NOAA, USGS > citation style is wrong; year is missing Author response: We will fix the citation style.

33) P4 L1 please describe differences between the two sites (pile 1 and 2), e.g. shadow, slope Author response: We will include differences between the piles within the Settings section.

34) L5 provide a reference to snow density section Author response: We will provide average snow density ranges for comparison.

35) L5 provide more information on the properties of the plastic sheets (e.g. thickness, size, water permeability, thermal conductivity . . .) and for what reason they were used (I guess to reduce snow pollution as stated later); such information should also be given for the foam used in for the insulation experiments Author response: Thank you for the suggestion. We will provide the properties we are aware of – you are correct that they were used to reduce snow contamination by woodchips. We will include a similar rationale and properties for the foam.

36) L6 brackets are missing (Fig. 3) Author response: We will include brackets.

37) L9 at which height above ground were the meteorological measurements performed? Author response: The measurements were preformed at a ~3m above ground. We will include this information.

38) L12 be more clear about soil temperatures: how many sensors? Where were the sensors? Where the sensors in the ground or in the covering layer? Author response: We will be more specific about the ground temperature sensor details in revisions.

39) L15 this section requires more details: the dates of the scans should be provided, e.g. in a table; Also add a table with the technical specifications of the laser scanner; Was multi-station adjustment used for registration; why not? It is an easy approach to improve registration of the data; What is the accuracy of the data? Were data gaps (scan shadows) existing? How were they handled? If a direct accuracy evaluation of the data is not possible, at least references to earlier studies that assessed TLS accuracy in similar settings should be added, e.g. Prokop et al. 2008, Grùl̇newald et al. 2010, Grùl̇newald and Wolfsperger 2019; Author response: Thanks for these suggestions and additional references on TLS accuracy. We will update the manuscript to provide further details on the workflow of scanning. We did run MSA in RiScan and will specify that detail. We did not conduct an analysis of data gaps, but given the relatively small size of the piles, it was feasible to get very good coverage with few scan positions – and thus had minimal data gaps. We will revise manuscript to clarify DEM generation settings about filling of any potential data gaps.

40) L32 please add for how long the insulation experiments lasted; until end of summer? Author response: These experiments lasted a week. It is possible to find this information in the accompanying figures; we will include it in the text as well.

41) L32 please state what kind of R (e.g. Pearson's correlation coefficient) is used Author response: We will be clear about what kind of "R" value we use.

42) P5 Sect 5.1 Sum of precipitation should also be given; How were condition of the recorded summer season in relation to long term climate? Data from station COC

described in Sect. 3 could be used to rate this summer; Author response: Great suggestion to place the conditions of summer 2018 into context – we will include this information.

43) L10-17: It is not clear which measurements are described here: the sensor below the piles or the ones next to the piles? Is there an explanation for the much larger T – variability for the 5 cm sensor at site 1 in relation to site 2? To which of the two sites does Fig 5 refer to? Author response: The measurements refer to the sensors next to the pile. The ground temperature data in Site 2 was disrupted halfway through the winter due to a faulty sensor and because of the frozen ground, we were unable to replace until warmer weather. Figure 5's panel a) is data from Site 1 – we will be sure to specify in the following draft. Data in panels b) - d) are from the weather station located closer to Site 1.

44) L17-19 unclear: only measurements of one site (below pile or next to pile) are shown in Fig 5; Author response: Yes – the only measurement displayed in Fig. 5 is the GT next to the pile in Site 2. We will specify.

45) L24: add a reference to Fig 3 (after . . ."for pile 2.") Author response: We will add this reference.

46) L25 use kg/m3 instead of g/m-3 Author response: Thank you for the suggestion – we will switch from g to kg.

47) L25-26 Where were densities measured (in which depth) obtained? Densification should be related and discussed in relation to the results of Grülnewald et al. 2018 who showed an increase in density, both in time and in depth; Author response: Density was collected at the top of one pile three times. We will be provide these details.

48) L26-27 "Relative to . . . (0.9g/cm-3)." Relating density to fresh snow is not meaningful in that context and could be removed; Author response: We believe this context is useful to demonstrate to diverse audiences the high level of snow compaction that

occurred over the summer. We will leave the context.

49) L27: I do not think that this is an adequate explanation. Snow with a density of 500 kg/m3 should already be fully decomposed and rounded; Was the snow dry during density measurements? Or was there some liquid water content? Or did you identify ice aggregations resulting from refrozen water? What was the grain size in March? Author response: Quantitatively analyzing snow morphology was not within the scope of this study, though we will include qualitative observations (wet as opposed to dry snow, high liquid water content).

50) L29: Please check the numbers: Considering the very similar melt rates of the two sites (Fig 7) the difference between 1.24 and 1.5 m3/d seems very high; is the removal of the 30m3 snow possible part of the melt rate? Author response: We will review our calculations to determine melt rate. The removal of snow halfway through the summer was not included in the calculations.

51) L29-32 Discussing melt rates is the main focus of the paper; Please discuss them in more detail; Your data set should allow a much more detailed analysis! e.g. how do melt rates change in time and how does this related to meteorology? Do melt rates vary spatially? What is the difference between the two piles? What is the difference between sections with different cover material? Author response: As described in comment 2, we will do further analysis of the spatial/temporal patterns of melt rate on the piles. While there are limitation to this analysis given the insulation depth was not constant (due to shifting/sliding) and the insulation experiments were preformed over a 1 m X 1 m section of the 200 $m^3$ piles, we believe it will still add further insight to melt processes.

52) L32-37 possibly even the effect of the crevasses could be seen in the TLS data (e.g. local changes in melt rates? Author response: Yes, crevasses are visible in the TLS data. However, there were so many factors influencing melt rate that we could not draw direct relationships between the forming of the crevasses and melt rate.

53) P6 Sect 5.2. This section is pretty poor. It should be enhanced: a discussion and reasoning on the effects of the different covering types (properties of materials and how do they interact with snow and atmosphere is missing; Currently only temperatures are analyzed but this is not enough to judge performance of the different materials; The TLS data could be used to quantify and discuss if and how volume losses differ under different covering materials. Author response: Thanks for the general suggestions for improving this section. There is some discussion of the material properties and interactions with atmosphere within the manuscript, but we can enhance the section.

54) add references to the specific panels for Fig 4 Author response: We will revise the manuscript to specifically reference individual panels of Figure 4.

55) L2 insulation efficiency is not only a function of T, e.g longwave emission or turbulent fluxes are not only depending to T but very relevant for the energy balance; Author response: Thank you for this note. We will incorporate this language into the paper, though we do not have longwave emission data.

56) L12 the presented experiments used wood chips and a plastic planked not only wood chips; Author response: You're correct – thanks for the clarification. We will make this change.

57) L13-14 climate is not only a function of latitude and elevation; please rephrase Author response: We will rephrase to be sure we're acknowledging that there are more factors that influence climate apart from latitude and elevation.

58) L 15 (fairbanksmuseum, 2019)) > remove bracket Author response: We will remove the bracket.

59) L36 Provide more details on the PSD method; how does it work and what is its benefit? How is it interpreted? Add references; Author response: The PSD section will be reworked to be clearer.

60) P7 L6-9 This explanation is too simple: heat transfer is not simply depending on air

temperature; surface temperature, cloudiness (longwave radiation) and wind (turbulent fluxes) are also crucial; See discussion of simulation results in Grǜl̇newald et al. 2018 and the sections about energy balance, and snow melt of the recent review paper of Mott et al. 2018; these references and possible also other earlier work should be cited in context of the discussion; Author response: Thanks for providing suggestions to strengthen this section – we will revise to acknowledge the different factors that influence heat transfer.

61) L11 what is the "R-value"? Author response: "R-value" is an accepted term in the United States refers to the insulating abilities of a material. We will fully explain "R-value" in the next revision.

62) Section 7: Conclusions should be prolonged; Here all three research questions form the introduction should be shortly answered; an outlook on future research that might be useful to enhance our understanding on snow storage might also be added; Author response: We agree that the conclusion should be restructured to include our three research questions and provide a clearer path for future research.

Figures 63) Figure 1 b) it would be nice if the list would be ordered geographically; Several sites are missing (see attached pdf; Reference: Wolfsperger et al 2018) Author response: Thanks for the geographic organization suggestion – we will reorganize in a more intuitive way. We appreciate the addition of the new sites.

64) Figure 4: T fluctuation of the blue line is hardly visible; possibly change axis or figure dimension Author response: The blue line actually does not fluctuate as it records temperature at the snow-insulation interface – which is likely why it is difficult to see it.

65) Figure 5: Figure a should be enlarged vertically to improve readability; grids or vertical lines should be added; For humidity and radiation adding daily mean values as line could also help to improve readability; Legend: To which snow pile does the figure refer to? Ground temperatures below or next to pile? Author response: We will enlarge the figure to the specifications allowed and we will improve readability. We will

be clearer about the origins of the GT values.

66) Figure 6: Please add a legend relating colors to dates. Author response: Thanks for the suggestion – we will include a legend.

67) Figure 7: Why is the increase in volume from April 1 to May 1 for site 2 so much larger than for site 1? Was there such a big difference in volume of chips added? Are colors between the two panels possibly mixed? The huge melt rate drop on July 1 might be correct for Site 1 (blue) but not for site 2; Add a grid or horizontal lines for readability; Author response: The difference in wood chips is partially the result of the shape of the pile – pile 1, which received less wood chips, was banked against the side of a hill while pile 2, which received slightly less than twice the amount of wood chips, was a domed pile. Because these piles had difference geometry and thus different surface areas, different amounts of wood chips were needed to cover them. The melt rate calculations will be checked and the graph will be improved for readability.

68) P9 L15 doi seems to be wrong L22 and L 24 The papers are not cited in the text; Having only checked few selected references I found three mistakes; I guess that there are more. Please check your citations and references carefully! Author response: Thanks for checking references. We will more carefully check and revise them.

Please also note the supplement to this comment:
https://www.the-cryosphere-discuss.net/tc-2019-56/tc-2019-56-AC3-supplement.pdf

---

## Referee Report (RR1)

The paper has improved significantly since the first issue. It is well written and I find objectives clear and the discussions and results now answers the presented research questions.

The paper presents novel research. Testing the optimized strategy from the two small experimental piles on a larger pile gave further value to the study. This paper complete other papers about snow storage and is valuable for both stakeholders dealing with snow storage as well as for future research on this subject.

I suggest the paper to be accepted for publication.

---

## Author Response (AR2)

Manuscript Responses to Reviewer

Author response in green.

5  Suggestions for revision or reasons for rejection (will be published if the paper is accepted for final publication)

The paper has been improved considerably and most of the suggestions of the first review round have been addressed. Moreover, and additional data set obtained for a larger pile during 2019 has
10  been added and confirms operational suitability of snow storage at the site.

What I am still missing are two points:
1) More details on spatial heterogeneity of snow melt at the piles
Author response: The spatial heterogeneity of snow melt of the 2018 data is not easily done, nor,
15  we believe, as important because of the small pile size (200 cubic meters). However, this analysis preformed on the more realistically sized 2019 pile provides critical observations (see new Fig. 9).

2) A comparison of measured melt for the different cover experiments.
Author response: We agree that this would be a useful analysis to include, however we do not
20  have the data resolution in time to support this analysis. While thermal data were collected at 5-minute intervals over the course of about a week for the cover experiments, the TLS data were collected every 10-14 days (on either side of the cover experiments but not always right before and right after). Additionally, the cover experiments took place on small sections of the two different piles (four 1 m x 1 m plots) instead of the entire pile which would make accurately
25  determining the rate of volume change for each individual pile unrepresentative. Working on two different piles would also make comparison difficult.

These supplements should be added and would further increase the impact of the study. Some more smaller (mainly technical comments) below.
30
P 1 L21 Add interval of the TLS measurements
Author response: We have made the change.

L21/22 If I understand correctly, in that context melt rate and snow volume change describe the
35  same process; The different nomenclature is a bit confusing and I suggest to consistently use volume change (melt rate usually is given in mm water equivalent).
Author response: We have changed "melt rate" to "rate of volume change" throughout the manuscript.

40  Please be also consistent with the sign before melt rates/ snow volume change (- or +).
Author response: We have changed the sign for consistency.

One other thing is that geometry, volume and surface of the two piles are not identical, it is therefore hard to compare their volume changes; scaling melt rate to surface area (or to initial
45  volume) would be a possible solution.
Author response: We agree that due to geometry, volume loss is not comparable. The 2018 piles had very similar volumes but the 2019 pile is 45 time larger. We have now included percentage loss per day values to account for this difference (2018 mean of 0.6 and 0.7% per day vs. 2019 mean of 0.16% per day).
50
L24 please also mention here which other type of cover material was tested
Author response: The cover materials tested are mentioned earlier in the abstract when discussing the cover experiments – line 24 specifically discusses just the covering strategy that we found most successful. We thought it redundant to mention the covers here again.
55
Fig 1: typo 19 Nordkette Innsbruck; Kitzbühel (Austria) would be another place to add (for alpine skiing); it should maybe be added that these places performed snow storage at least once but that they are not necessarily still operating (e.g. Sochi was only for the Olympics…)
Author response: Thank you for the suggestions. They have been incorporated.

P3 L1 Short wave radiation also increases significantly with elevation
Author response: The change has been incorporated.

P4 L31 remove ) > (Riegel VZ 1000))
Author response: The change has been incorporated.

L31 in that context "digital surface model" would be the correct nomenclature
Author response: The change has been incorporated.

P5 L31 provide a reference for Riscan pro
Author response: This seems to be a mislabeled line number since pg. 5 line 31 does not involved the RiSCAN Pro software. We did include a reference to the RiSCAN pro manual and then cited it when we discussed the software.

Section 3.6 I think it would be good to include a table illustrating the different settings and properties of the cover experiments
Author response: The table has been included.

L 25 why was the plastic layer removed for the experiment with wood chips? What about the reflective cover?
Author response: The plastic layer was removed for the wood chip experiment to test effectiveness of wood chips alone against snow. The reflective cover was tested to determine the effectiveness of re-radiating short-wave radiation. We removed the plastic because we suspected it contributed to plie failure by channeling rain water into the snow as concentrated flow.

P6 L10 No plastic foil at the base for that experiment?
Author response: Correct - there was no plastic at the base for the experiment.

L 14 write TLS instead of Lidar – no reason to introduce an additional abbreviation
Author response: The change has been incorporated.

P7 Sect 4.2. Still missing is an analysis/discussion of spatial variability in melt rates of the two piles. Figure 6 shows large spatial and temporal variability. But it is hardly analysed in the text…
Author response: We will include information about the spatial variability of volume loss for the 2019 pile (see new Fig. 9).

Figure 6: Please add a North-arrow as an indication of slope expositions.
Author response: A North arrow has been included.

P7 L33 How can you explain the huge temperature rise at the end of the period for panes c?
Author response: We pulled the temperature probes out at that point which exposed all of them to direct solar radiation. We have edited the graph to not include those points.

The rise to 23.8°C for panel e is not visible in the figure. Is it correct?
Author response: As this information was not adding to the paper, it has been deleted.

What is the measurement accuracy/precision of the temperature loggers? Is it really 0.01°C? If not I suggest to round numbers to less digits.
Author response: According to the HOBO website It is an accuracy of 0.5 C, however, as this information was not adding to the paper, it has been deleted.

As already suggested in the first review, I still think it would be worth to analyse and compare snow-depth change of the different cover experiments. It would be interesting if these direct measurements of melt confirm you findings from the temperature loggers. Of course, the areas are quite small but it is worth a try.
Author response: As stated above, we agree that this would be an impactful comparison if we had data for it. However, our data do not allow us to analyze volume change this way. Our experimental plots are too small (1 m X 1 m) in comparison with our 200 cubic meter pile and the

experiments' data collection took place between two scans – there were not scans taken while the experiment was occurring nor right before and right after.

Sect. 4.6 Many readers (including myself) would benefit from some more assistance in interpretation of Figure 8. I think you should add a sentence describing in more detail, how frequency in the figure is translated to periodicy (24 h, 12h ….).
Author Response: We have included more information about Fig. 8.

For illustration, these peaks could be marked e.g with arrows in the Figure. Moreover, it is a bit confusing that Fig 8b refers to 4f and 8c to 4e – why not keep the order of Fig 4?
Author response: Both these suggestions have been incorporated.

One should also state more clearly, which covering method performed best. This is one of the most important findings of your study. If I understand correctly the order would be 8c -8a- 8b;
Author response: We have incorporated this ranking.

P8 L9/10: It might also be an option to add an axis to Fig 8, which already shows the translation)
Author response: We have chosen not to add the axis to maintain simplicity and readability of the graph.

Fig 8: add a label "depth of T-sensor" to the legends
Author response: Labels have been added but not depth, rather height about snow.

Fig 8: From the legend and from the caption it is not clear which sensor belongs to which curve: caption: "depth in cm measured below uppermost sensor" - this means that 0 cm should be the uppermost sensor but the bold purple line is the "on snow" sensor if I understand correctly and if I compare to Fig 4…
Author response: We have reworded to alleviate confusion.

L15 You should relate % daily loss for 2019 to 2018; Even though not the same year, the similar climatic conditions allow for a qualitative comparison and dare confirming the importance of storage volume for loss (as partly discussed in Sect 5.1).

Author response: We have included percentage loss per day values which allow comparison between both years in section 5.3 when we discuss data from 2019.

P8 L 22,25,29 TLS instead of Lidar –also later in the text
Author response: The change has been made.

P9 L 8: you should state here that the 2019-pile confirmed the suitability of snow storage at your place.
Author response: The line has been added.

Sect 5.2 When talking about the most effective covering you should also consider and discuss the different costs. Cover materials have different prices and life spans, work load might vary considerably. If you use more cover material (e.g. deeper layers of saw-dust or additional covers) you will increase isolation (as also shown in Grünewald et al. 2018) but also costs. The optimal strategy is somewhere in-between. In the Conclusions you mention briefly that the cover material was changed for 2019. But more details would be appreciated.
Author response: We agree that cost is an important piece to mention and we have included a couple sentences to address this issue; however, a more complete financial analysis is not within the scope of this paper.

Sect 5.3 The experiments of 2018 showed that reflective blanket+wood chips + concrete blanket performed best. In 2019 wood chips and a different blanket was used. It would be good to summarize here why the setting was changed – why not concrete blanket….?
Author response: The concrete blanket was not used because of cost (per above comment) and logistical concerns of the COC. We now include a sentence to acknowledge this. A permeable rather than impermeable cover was used to avoid run off and associated regulatory concerns.

Sect. 6: Some outlook on questions that remain open for further research on snow-storage might be added.

Author response: Thank you for the suggestion. Some further research ideas were included.

Relevant Changes in the Manuscript:
- Technical changes were made per the reviewer's suggestions
- A spatial variability analysis was included on the 2019 snow pile (Fig. 9)
- The Power Spectral Density section (4.6) was added to and revised

[revised manuscript text omitted]

**Page 28: [3] Deleted**             **Hannah Weiss**             **10/11/19 1:18:00 PM**

**Page 28: [3] Deleted**             **Hannah Weiss**             **10/11/19 1:18:00 PM**

**Page 28: [3] Deleted**             **Hannah Weiss**             **10/11/19 1:18:00 PM**

**Page 28: [3] Deleted**             **Hannah Weiss**             **10/11/19 1:18:00 PM**

**Page 28: [3] Deleted**             **Hannah Weiss**             **10/11/19 1:18:00 PM**

**Page 28: [3] Deleted**             **Hannah Weiss**             **10/11/19 1:18:00 PM**

**Page 28: [3] Deleted**             **Hannah Weiss**             **10/11/19 1:18:00 PM**

**Page 28: [3] Deleted**             **Hannah Weiss**             **10/11/19 1:18:00 PM**

**Page 28: [3] Deleted**             **Hannah Weiss**             **10/11/19 1:18:00 PM**